



# Modelling hourly evapotranspiration in urban environments with SCOPE using open remote sensing and meteorological data

Alby Duarte Rocha[1], Stenka Vulova[1], Christiaan van der Tol[2], Michael Förster[1], Birgit Kleinschmit[1]

[1] Geoinformation in Environmental Planning Lab, Technische Universität Berlin, 10623 Berlin, Germany

[2] University of Twente, Faculty of Geo-Information Science and Earth Observation (ITC), P.O. Box 217, Enschede AE7500, The Netherlands

*Correspondence to*: Alby Duarte Rocha (a.duarterocha@tu-berlin.de)

**Abstract.** Evapotranspiration (ET) is a fundamental variable to assess water balance and urban heat island effect. ET is deeply dependent on the land cover as it derives mainly from the processes of soil evaporation and plant transpiration. The majority

of well-known process-based models based on the Penman-Monteith equation focus on the atmospheric interfaces (e.g. radiation, temperature and humidity), lacking explicit input parameters to describe the land surface. The model Soil-Canopy-Observation of Photosynthesis and Energy fluxes (SCOPE) accounts for a broad range of surface-atmosphere interactions to predict ET. However, like most modelling approaches, SCOPE assumes a homogeneous vegetated landscape to estimate ET. Urban environments are highly fragmented, exhibiting a blend of pervious and impervious anthropogenic surfaces. Whereas,

high-resolution remote sensing (RS) and detailed GIS information to characterise land surfaces is usually available for major cities. Data describing land surface properties were used in this study to develop a method to correct bias in ET predictions caused by the assumption of homogeneous vegetation by process-based models. Two urban sites equipped with eddy flux towers presenting different levels of vegetation fraction and imperviousness located in Berlin, Germany, were used as study cases. The correction factor for urban environments has increased model accuracy significantly, reducing the relative bias in

ET predictions from 0.74 to -0.001 and 2.20 to -0.13 for the two sites, respectively, considering the SCOPE model using RS data. Model errors (i.e. RMSE) were also considerably reduced in both sites, from 0.061 to 0.026 and 0.100 to 0.021, while the coefficient of determination (R2) remained similar after the correction, 0.82 and 0.47, respectively. This study presents a novel method to predict hourly urban ET using freely available RS and meteorological data, independently from the flux tower measurements. The presented method can support actions to mitigate climate change in urban areas, where most the world

population lives.

## 1 Introduction

Evapotranspiration (ET) is essential for understanding water cycle and energy balance, regulating precipitation, temperature and vegetation productivity (Wang et al., 2020; Zheng et al., 2020). In urban environments, ET is inversely proportional to the intensity of the urban heat island (UHI) effect (Wang et al., 2020). The UHI effect adversely affects the health and quality of

life of urban residents (Kovats and Hajat, 2008; Scherer et al., 2013; Vulova et al., 2020). As most of the world population



lives in cities (United Nations, 2019) and climate change is expected to further increase the frequency and intensity of heat islands (Huang et al., 2019), actions to mitigate the UHI effect are essential. Optimising ET capacity in urban areas could reduce extreme events such as severe heat waves, drought or flooding (Wang et al., 2020; Ward and Grimmond, 2017). Although ET is undeniably important for planning more sustainable cities, ET studies in urban environments are rare due to

the challenges of measuring and modelling it in a highly heterogeneous landscape (Nouri et al., 2015, 2020).

ET is a measurement of mass (millimetres) or energy (watts or joules) of the movement of water from the land surface to the atmosphere (i.e. liquid to vapour) (Liang and Wang, 2020). Terrestrial ET is the sum of three main sources of evaporation from the land surface: a) evaporation from soil moisture and groundwater; b) evaporation from plant transpiration; and c) evaporation from intercepted precipitation (Miralles et al., 2020; Nouri et al., 2019). Climatology, hydrology and agriculture

have developed different instruments and methods to estimate ET (Nouri et al., 2013). ET values can be derived from micrometeorological approaches using instruments such as eddy flux towers (atmosphere), hydrological approaches by lysimeters (soil), or physiological approaches such as sap flow or chamber systems (vegetation). These instruments can provide quite different estimates as soil evaporation, transpiration and interception loss are very distinctive processes which are challenging to separate, particularly when measured by micrometeorological approaches (Miralles et al., 2020).

ET is mainly driven by atmospheric conditions such as sunlight intensity (i.e. incoming radiation), air temperature and relative humidity (Foltýnová et al., 2020). In contrast, the volume of ET over the year depends greatly on the land surface and the water availability in the soil (Dwarakish et al., 2015; Wang et al., 2020; Zheng et al., 2020). In climatology, ET is often measured by the eddy covariance method, which is based on the turbulence flux and energy balance (Liang and Wang, 2020). This method is affected by atmospheric stability, wind profile and surface roughness in the surroundings of the flux tower

(Foltýnová et al., 2020; Schmid and Oke, 1990; Ward and Grimmond, 2017). The ET observations are continually collected over regular time intervals but represented by an irregular area based on footprints which change in shape, size and orientation according to atmospheric conditions (Kljun et al., 2002; Kotthaus and Grimmond, 2014).

Traditional point-based techniques to measure ET are insufficient in a heterogeneous urban environment (Nouri et al., 2013). ET observations collected by lysimeters are affected by percolation, runoff, soil moisture and the volume of the ground table

water (Nouri et al., 2019). Being measured by installation under the ground, the measured area is very limited in extent and difficult to compare to the surrounding areas in urban environments (Nouri et al., 2013). Furthermore, instruments such as gas exchange chambers for leaves or sap flow for individual trees only measure transpiration in plants rather than total ET (Kuhlemann et al., 2020; Maltese et al., 2018). None of these techniques are well-suited for the urban environment, whereas eddy covariance (EC) is the most adequate system to directly measure ET in cities (Liang and Wang, 2020; Nouri et al., 2013).

However, eddy covariance measurements represent a relatively small and constantly varying land cover area around the flux tower (diameter of ~500m), making it impractical to set up a widespread network of flux towers over the city given the high costs to install and operate it (Westerhoff, 2015).

A practical solution is to estimate ET using process-based or empirical models. Fitting classical empirical models or machine learning algorithms using meteorological inputs and spectral indices is relatively common in natural landscapes, but relatively





rare in an urban environment (Vulova et al., 2021; Wang et al., 2020). One reason is the necessity to train the model at representative locations and conditions, which is a challenge in urban areas due the constantly changing land cover captured by the tower's footprint and lack of flux towers at different surfaces in a highly fragmented and heterogeneous environment (Feigenwinter et al., 2018). In addition, most of the widely used empirical models are unsuitable for variables with strong spatiotemporal dependency such as ET (Rocha et al., 2018, 2020).

Hydrological models such as MIKE SHE, Variable Infiltration Capacity (VIC) and Soil and Water Assessment Tool (SWAT) are process-based models focus on streamflow, soil moisture storage and runoff generation processes which can provide estimations of vegetation transpiration, soil evaporation, and interception loss (Dwarakish et al., 2015; Zhao et al., 2013). The ET estimations from hydrological models are often based on energy balance and mass transfer methods such as the Penman-Monteith equation (Devia et al., 2015; Zhao et al., 2013). Penman-Monteith equations are widely used in literature, especially

for agricultural applications (Allen et al., 2005). However, this approach focuses mostly on the atmospheric interface and stomatal conductance. The original formulation of the Penman-Monteith equation uses simplified assumptions of land cover and growth, lacking the capacity to capture the variation in land surface or plant phenology (Westerhoff, 2015). To overcome this limitation, it is common to include remote sensing data such as the normalized difference vegetation index (NDVI) as a proxy of leaf area index (LAI) or biomass (Boegh et al., 2004; van der Tol and Norberto, 2012; Westerhoff, 2015).

Surface Energy Balance (SEB) model versions such as Surface Energy Balance Algorithm for Land (SEBAL), Surface energy balance index (SEBI) and Surface Energy Balance System (SEBS) include variables retrieved from remote sensing such as land surface temperature, albedo, and net radiation, but still require meteorological data (Bayat et al., 2018; Nouri et al., 2015; van der Tol and Norberto, 2012). SEBAL models which include RS derived parameters, such as LAI, are therefore semi-empirical and spatially dependent, reducing the capacity to generalise to other locations (Rocha et al., 2020). Although many

input variables for modelling ET are currently derived from optical and thermal sensors, ET cannot be empirically derived exclusively from remote sensing as it does not directly change the spectral reflectance (Timmermans et al., 2013; van der Tol and Norberto, 2012).

Soil-Vegetation-Atmosphere Transfer (SVAT) models are also based on energy balance and mass transfer but allow for more comprehensive parameterisation of soil and vegetation surface properties (Kracher et al., 2009; Petropoulos et al., 2009). The

Soil-Canopy- Observation of Photosynthesis and Energy fluxes (SCOPE) is a SVAT model that accounts for surface-atmosphere interactions of both turbulent heat fluxes and radiative transfer (van der Tol et al., 2009). SCOPE has been successfully applied to predict ET in croplands and natural environments (Bayat et al., 2018, 2019; Timmermans et al., 2013) and the sum of latent and sensible heat flux (Duffour et al., 2015).

Most ET modelling approaches assume a landscape of homogeneous vegetation without anthropogenic elements (Nouri et al.,

2015). On the one hand, urban environments are highly fragmented and present heterogeneous pervious and impervious surfaces (vertically and horizontally) (Feigenwinter et al., 2012; Ward and Grimmond, 2017; Zheng et al., 2020). The effect of surface heterogeneity in the horizontal direction, typical in an urban environment, is not addressed by (1D) models such as SCOPE or Penman-Monteith-based models (van der Tol et al., 2009; Yang et al., 2020). On the other hand, the urban




environment offers the opportunity to integrate information from remote sensing (RS) and GIS at a fine spatial resolution to
represent the variation in land cover and surface properties (Boegh et al., 2009; Nouri et al., 2020).

The aim of this study is to develop a robust method to map evaporation in an urban area, by combining process-based models
(i.e. SCOPE), meteorological measurements, and remote sensing data. In an urban environment partially vegetated, to what
extent SCOPE model and detailed land surface information can provide accurate estimates of ET compared with eddy
covariance flux tower measurements. The method consists in correct the output of the 1-D SCOPE model by the subtraction
of the impervious areas using footprint modelling.

The main goals of this study are 1) to predict ET using input variables from publicly available data from standard
meteorological stations and remote sensing while assessing the accuracy independently with ET measured by flux towers at
two site locations; 2) to predict hourly ET for an entire year (12 months, 24 hours) using the SCOPE model in comparison
with reference ET derived from the Penman-Monteith equation; 3) to correct model bias due to the fragmented vegetation
cover and impervious surfaces using a factor derived from RS/GIS data extracted by hourly footprints. The novelties of this
study are that, to the best of our knowledge, the SCOPE model was never applied to urban areas and in the development of a
factor to correct prediction bias in urban environments using RS/GIS information.

## 2 Methods

### 2.1 Study area

Two sites in Germany's biggest city and capital, Berlin, were selected for this study because they are equipped with flux towers
using similar eddy covariance instrumentation but located in areas with differing levels of vegetation cover and
imperviousness. One site is located in a relatively green neighbourhood, while the other site is in a central built-up area. The
two sites are 6 km far apart from each other (Fig.1).

Berlin is situated in a temperate climate zone with humid sea air, presenting mild temperatures when air masses come from
southerly directions and cooler air from the (Atlantic) north (Senate Department for Urban Planning and the Environment,
2015). Easterly air masses or continental wind directions usually bring extremely dry air and may cause very cold periods in
winter and exceptionally hot days in summer. Berlin is mainly flat with an elevation of 34 meters above the sea (from 24 m to
120 m). The maximum volume of the annual precipitation (mm) occurs in the summer, whereas winter months present the
highest number of hours under rainfall. The lowest precipitation (volume and occurrences) is often in the transitional seasons,
with the driest month usually being April (Fig. 2).



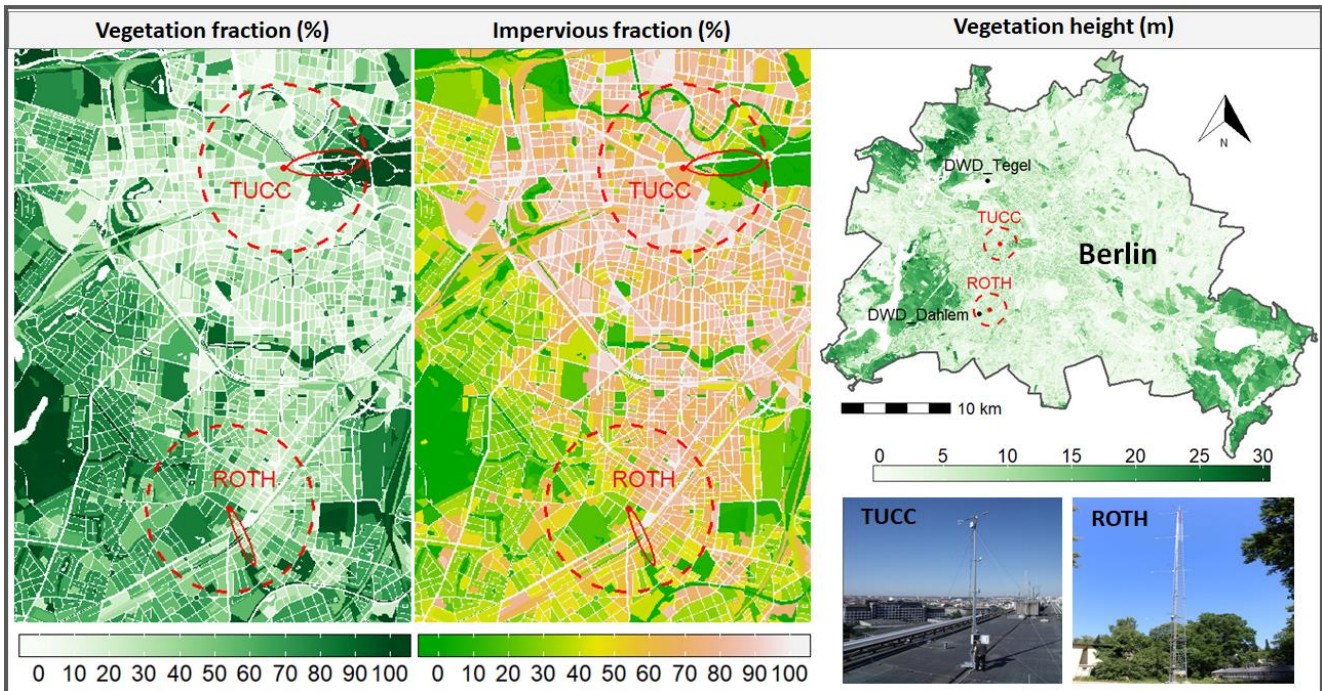

**Figure 1.** Locations of the two sites with the respective vegetation fraction (%), impervious fraction (%) and vegetation height (m) in the surroundings of the flux towers. The red dotted areas represent a buffer of 1500 m around the towers (red dot), while the red ellipse are examples of footprints for a 30 minutes timestamp in the year of 2019. The black dots on the Berlin vegetation height map refer to the DWD stations Tegel and Dahlem. The three land surface maps were extracted from the Berlin Digital Environmental Atlas (Senate Department for Urban Development and Housing, 2017b, 2017a; Senate Department for Urban Planning and the Environment, 2014).

## 2.2 Data

### 2.2.1 Eddy covariance flux towers

The two eddy covariance flux towers are operated by the Chair of Climatology at the Technische Universität Berlin (TUB) as part of the Urban Climate Observatory (UCO) (Scherer et al., 2019; Vulova et al., 2021). The flux towers measure turbulent fluxes of sensible and latent heat based on the EC method. The EC systems operated by UCO include an open-path gas analyser and a three-dimensional sonic anemometer-thermometer (IRGASON, Campbell Scientific). The towers are equipped with micrometeorological instruments to simultaneously measure water vapour density and orthogonal wind components, allowing for ET estimation.

The flux tower referred to as Rothenburgstraße (ROTH) is located in a research garden in the southwest of the city. The observations at ROTH are measured at approximately 40 m height above the ground, a few meters higher than the tree canopies and the one building nearby. The other flux tower, called TUB Campus Charlottenburg (TUCC), is located on top of the main building of the university in the city centre (Fig. 1). The measurements are taken from a 10 m tower above the roof, for a total measurement height of 56 meters above the ground.



The complete time series of the year 2019 was selected for this study based on the simultaneous availability of data from both

towers. Data from winter months and at night time were included. Negative values (condensation) were set to zero. The data

were corrected to air density and sonic temperature for humidity, high- and low-frequency spectral corrections using double

coordinate rotation (Moncrieff et al., 1997; Webb et al., 1980). The 30-minute resolution time series was assessed based on

the values of latent heat flux (LE, W/m$^2$). LE values under the following conditions were excluded: (1) observations with flag

quality higher than one (Foken, 2008); (2) values outside of the thresholds of -100 W/m$^2$ and 500 W/m$^2$; (3) observations six

standard deviations (SD) greater than the average (de-spiking), and (4) measurements during precipitation or up to 4 hours

after rain events.  Items one to three were performed using functions from the R package "FREddyPro" (Xenakis, 2016).

The upward latent heat flux (LE, W/m$^2$) observations were aggregated to hourly resolution and converted to ET by the

expression ET = LE/$\lambda$, where $\lambda$ is the latent heat of vaporization (J kg$^{-1}$). ET was calculated as a function of air temperature

using the "bigleaf" R package (Knauer et al., 2018). The following measurements from both towers were used to calculate the

footprints: wind speed (ws, m s$^{-1}$), wind direction (wd, degree), friction velocity (u*, m s$^{-1}$), Obukhov length (L, m) and

northward wind (v_var, m$^2$ s$^{-2}$). After pre-processing, from the 8760 timestamps, 43 % of the ROTH and 42 % of the TUCC

data were missing. The remaining values of ET, 4993 and 5104 values respectively, were used to assess the model accuracy.

To obtain monthly and yearly ET estimates, gap-filling is required.  Given the strong seasonal and diurnal variation of ET,

linear interpolation is not recommended. A standard procedure is to use the marginal distribution sampling (MDS) gap-filling

algorithm, which considers meteorological variables to account for the daily and annual seasonality (Falge et al., 2001;

Foltýnová et al., 2020; Reichstein et al., 2005; Wutzler et al., 2018). The monthly and yearly values of ET from MDS gap-

filling will later be compared with the modelled ET predictions.

### 2.2.2 DWD meteorological data

In order to use model inputs completely independent from the flux towers, data from the meteorological stations of the German

Meteorological Service network (DWD Climate Data Center) were selected based on the distance to the flux towers (DWD,

2020). From the stations, Tegel (~5 km from TUCC) and Berlin-Dahlem (~1 km from ROTH), hourly time series of air

temperature, air pressure, relative humidity, wind speed, wind direction, and precipitation (occurrence and volume) were

collected. The variables shortwave and longwave radiation were collected from the Potsdam station to represent both sites.

Potsdam station is located in the neighbouring city with the same name, ~19 km and ~23 km from the flux tower sites.

### 2.2.3 Remote sensing and GIS data

The LAI300m (V1) product generated by the Global Land Service of Copernicus, the Earth Observation program of the

European Commission, provides a valuable estimate of an essential biophysical parameter to model ET (Table 1). The

Copernicus product provides a grid of LAI values with 333 meters spatial resolution and ten days temporal resolution (Bauer-

Marschallinger and Paulik, 2019). The product is based on PROBA-V data and the LAI was estimated by neural network

algorithms trained with MODIS and CYCLOPES products. The product was atmospherically corrected, with outlier's removal





and cloud masking. Smoothing and gap-filling operations were applied based on the land cover type and temporal performance. A time series of 36 LAI maps for 2019 were downloaded and linearly interpolated to match the timestamp of observed ET. We assumed that in between the 10-days gap, the differences that occur were relatively minor and irrelevant to this study.


**Table 1.** Datasets and data sources used to model ET in this study

| Dataset | Variables | Sources |
|---|---|---|
| Meteorological data -DWD stations | Air temperature (Ta, ºC), air pressure (p, hPa), relative humidity (rH, %), wind speed (ws, m s$^{-1}$) and direction (wd, degree), precipitation events (Oc_prec, yes/no), precipitation volume (V_prec, mm/h), incoming shortwave radiation[a] (Rin, J/cm$^2$), incoming longwave radiation[a] (Rli, J/cm$^2$). | DWD Climate Data Center http://ftp-cdc.dwd.de/climate_environment/CDC/observations_germany/climate/hourly/ |
| Eddy covariance data - EC flux tower | Latent heat flux (LE, W m$^{-2}$), wind speed (ws, m s$^{-1}$), wind direction (wd, degree), friction velocity (u*, m s$^{-1}$), Obukhov length (L, m) and northward wind (v_var, m$^2$ s$^{-2}$). | Urban Climate Observatory (UCO); Chair of Climatology - Technische Universität Berlin (TUB) |
| Remote sensing data - Copernicus | Leaf Area Index - 300m resolution (LAI, unitless) | Global Land Service of Copernicus – Portal Distribution (http://land.copernicus.vgt.vito.be/PDF/portal/Application.html) |
| RS hyperspectral data - Soil samples | Soil spectral reflectance (Soil_ref, unitless) | It was collected using a field spectrometer (ASD3) with a probe at the ROTH site. |
| GIS/RS data - Berlin Environmental Atlas, Green Volume (Edition 2017) | Vegetation fraction (Veg_frac, %), vegetation height (hc, m), roughness length[b] (zo, m) and zero-plane displacement[b] (d, m) | Berlin Senate Department for Urban Development and Housing https://fbinter.stadt-berlin.de/fb/wfs/data/senstadt/s_05_09_gruenvol2010 |

(a) Rin and Rli were later transformed to [W m$^{-2}$]; and (b) calculated based on the vegetation and building height.

Despite the GIS data representing a surface in a specific period (fixed in time), the corresponding area of the EC flux

measurements (e.g. ET or LE) continually varies in shape, size and orientation. Therefore, areas of influence (footprints) were



calculated for every half-hourly of 2019 for both towers to capture the spatiotemporal dynamics of the surface properties. The Kormann and Meixner (2001) analytical footprint model was applied using the R package "FREddyPro" (Xenakis, 2016). The footprints vary according to atmospheric stability (Monin-Obukhov stability parameter) and wind components (direction, speed, cross-stream, friction velocity) interacting with the surface roughness around the tower (i.e. 1500 m fetch size) (Kljun
et al., 2002; Kormann and Meixner, 2001).

The input data for footprint modelling were derived from the flux towers measurements, except for the aerodynamic parameter (zd), roughness length (zo) and zero-plane displacement (d), which were calculated from building and vegetation height by seasons (i.e. winter, summer, and intermediate) to incorporate changes in tree foliage. For further information about how the parameters were calculated, see Kent et al. (2017) and Quanz (2018). The footprints were based on a regular grid with 10m
resolution. Each pixel of the footprint is a probability which represents the chance of that area influencing flux measurements at a specific time (Schmid and Oke, 1990). Pixels out of the footprint area representing 90 % likelihood were considered as zero, and all non-zero pixels were rescaled to sum to a probability of one.

Surface properties to characterise the two Berlin sites were derived from a publicly available GIS database. Vegetation fraction (%) and vegetation height (m) was obtained from the Green Volume publication (edition 2017) from the Berlin Digital
Environmental Atlas (Senate Department for Urban Development and Housing, 2017a). All the layers of GIS maps were converted to a raster with 10 meters resolution and resampled to the grid used for calculating the footprints for each tower.

The raster layers of each land surface were then multiplied by a footprint raster and the resulting pixel values were summed to obtain the weighted averages of surface properties for each site and timestamp. The average vegetation and impervious fraction extracted from the footprints are relative values that vary from 0 to 1. In this study, water bodies were omitted as they represent
only 2.7 % of land cover at TUCC site and 0.0 % at ROTH on average. The Berlin Environmental Atlas also presents a detailed set of maps from the study "Surface runoff, percolation, total runoff and evaporation from precipitation" (Senate Department for Urban Planning and the Environment, 2019). This study will be used for comparison with our results.



**Figure 2.** Time series of the main variables used in this study for both sites in 2019, where green lines represent the data from the ROTH site and blue the TUCC site. (a) Air temperature (Ta), the dotted lines represent the maximum and minimum daily values and solid lines represent average daily values; (b) Incoming shortwave radiation (Rin) is common for both sites, where the solid black line represents the average and the dotted the maximum daily values; (c) LAI RS-derived values; (d) the volume of precipitation (mm); and (e) the evapotranspiration observations from the EC towers (ET).



### 2.3 Model approaches

#### 2.3.1 Penman-Monteith model

A formulation based on the Penman-Monteith equation (the ASCE standardised equation for short crops) was used to calculate reference ET (ETo) (Allen et al., 2005). Hourly ETo was calculated by providing air temperature, wind speed, relative humidity, and incoming shortwave radiation as model input using the R package "water" (Olmedo et al., 2016). As this formulation of ETo assumes a homogeneous landscape of short crops, no land surface information is required and the model is exclusively driven by meteorological conditions (table 2). Penman-Monteith ETo is a well-known and established approach which will be used as a baseline to evaluate to what extent including inputs that characterise surface properties can improve ET prediction accuracy.

#### 2.3.2 SCOPE model

The Soil-Canopy-Observation of Photosynthesis and Energy fluxes (SCOPE) is a process-based model (i.e. SVAT model), which integrates radiative transfer models (RTM) of soil, leaf and canopy with energy balance models (van der Tol et al., 2009). SCOPE is an ensemble model based combining traditional turbid medium radiative transfer, micrometeorology and plant physiology (van der Tol et al., 2009). This configuration allows SCOPE to account for a wide range of surface-atmosphere interactions, providing a vast combination of model inputs and outputs. Since SCOPE is a 1-D vertical model which assumes homogeneity in a horizontal direction, it is not designed for heterogeneous urban areas (Yang et al., 2020). Although LE cannot be directly retrieved from reflectance spectra, it can be estimated in the SCOPE forward model. The most important group of variables to estimate LE are (1) meteorological inputs such as incoming shortwave radiation (Rin), air temperature (Ta) and atmospheric vapour pressure (ea); (2) biochemical plant traits inputs such as the Ball-Berry stomatal conductance parameter (m) and maximum carboxylation capacity (Vcmax); and (3) biophysical inputs as leaf angle distribution (LIDFa, LIDFb) and LAI (Yang et al., 2020).

A list of the model inputs used in this study which varies across the timestamp is provided in Table 2. Since varying all model inputs of SCOPE realistically for a time series is unfeasible, the other parameters were kept constant. The other parameters were kept as the default in most cases, except for the roughness length (zo) and zero-plane displacement (d), which were set based on the footprints. Three scenarios were tested: (1) a SCOPE model with the same input variables as used for reference ETo (Penman-Monteith); (2) a SCOPE model with all available inputs from the DWD datasets; and (3) a SCOPE model that combines DWD data with RS data. The model output, total LE (W/m$^2$), was converted to ET (mm/hour) using the same procedure used with EC tower data. The modelling was performed in MATLAB R2018b using SCOPE version 2.0 (Yang et al., 2020).






**Table 2.** Input parameters which vary hourly for each SCOPE scenario

| Model inputs | SCOPE scenarios | | |
|---|---|---|---|
| | ETo | DWD | DWD+RS |
| Air temperature [ºC] (Ta) | X | X | X |
| Relative Humidity [ . ] (RH) | X | X | X |
| Wind speed [m s$^{-1}$] (u) | X | X | X |
| Incoming shortwave radiation [W m$^{-2}$] (Rin) | X | X | X |
| Incoming longwave radiation [W m$^{-2}$] (Rli) | | X | X |
| Air pressure [ppm] (p) | | X | X |
| Solar zenith angle [deg] (tts) | | X | X |
| Leaf Area Index [ . ] (LAI) | | | X |
| Vegetation height [m] (hc) | | | X |
| Soil reflectance [ . ] (soil_refl) | | | X |

(tts) was derived from the DWD timestamp. The setting options 'soil heat method' and 'applTcorr' to correct vcmax parameter by temperature were used to run the model scenarios DWD and RS.

**2.3.3 Correction factor for urban environments**

The vegetation fraction values extracted from footprints show considerable differences between the two sites (Fig. 1), while the meteorological data are quite similar (Fig. 2). The ROTH site presents an annual (footprint) average vegetation fraction of 49 % and canopy height of 5.5 m. In contrast, the TUCC site presents 27 % and 2.6 m, respectively. With 72 % of imperviousness, TUCC site is a denser built-up area compared to the ROTH site, with 49 %. As the models assume
homogeneous vegetation in the horizontal direction, ET predictions are likely to be biased if imperviousness is not accounted for.

In order to correct the predicted values according to the surface properties for each site, we propose a factor to correct the model output using the vegetation fraction extracted from the GIS layer by the hourly footprint. The correction factor for each timestamp is multiplied by the total ET predictions from SCOPE and ETo from Penman-Monteith. Then, corrected ET model
predictions are compared with the observed (hourly aggregated) ET from the flux tower to assess model accuracy.





### 2.3.4 Model assessment

As both models are fully deterministic, no train and test splitting or cross-validation approaches are needed to select and validate the models. The model accuracy was assessed using all available ET values from the flux tower time series. To assess model precision, the metrics Root Mean Square Error (RMSE) and the coefficient of determination ($R^2$) between predicted

and observed ET were used. Since deterministic or process-based models are more prone to prediction biases than fitted empirical models, a metric to assess relative bias (rBias) was used. In this study, rBias was also used as an indicator of the correction factor efficiency in providing unbiased predictions in an urban environment. All plots and metrics for model assessment were performed using the "ggplot2" package (Wickham, 2016) and basic functions in R software (R Core Team, 2020).

## 3. Results

### 3.1 ET prediction in urban environments

Process-based models derived from the Penman-Monteith equation focus mostly on the atmospheric conditions as these are the main drivers for evapotranspiration (Allen et al., 2005). In this study, we tested the similarity of the atmospheric conditions measured by the flux towers against nearby standard meteorological stations (DWD). The results show (Fig. 3a) that the

correlation between the ETo calculated from the two flux towers or two DWD stations is around 0.97 for any combination. This result indicates that the local atmospheric conditions can be represented by a nearby meteorological station without losing significant accuracy. Therefore, ET was predicted using independent input variables from publicly available meteorological stations (DWD) and the model accuracy assessed with ET observed from flux towers (EC method).



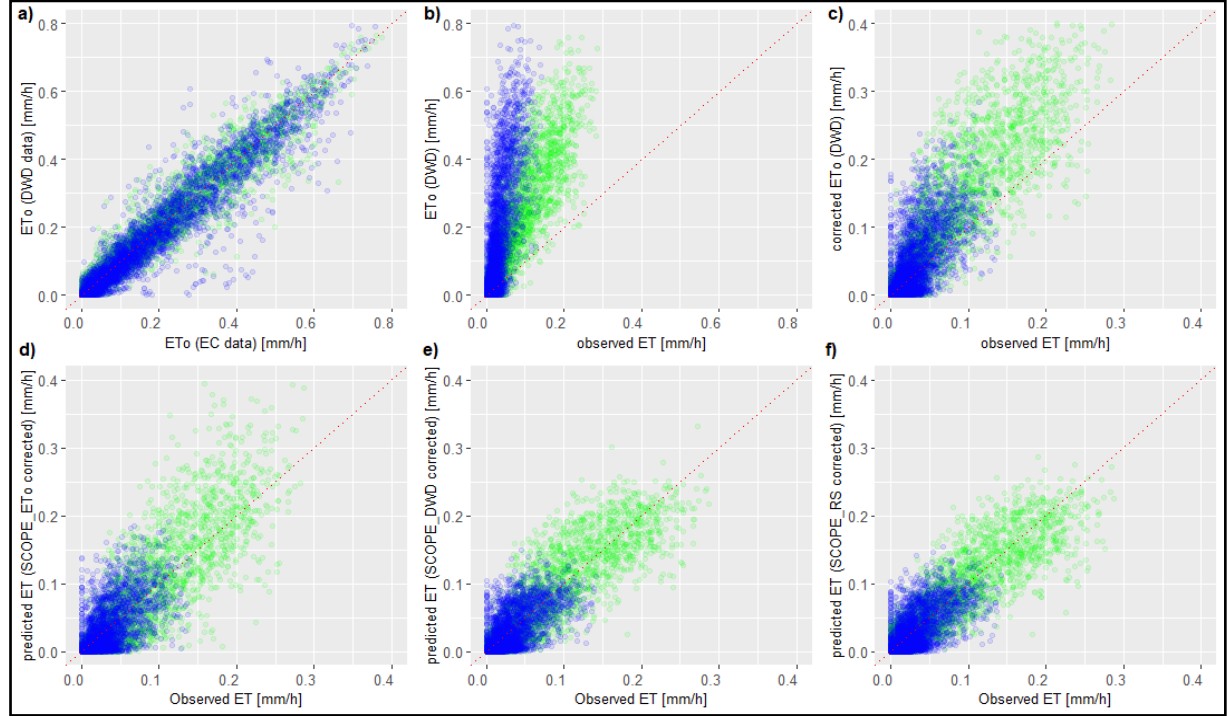

**Figure 3.** The relationship between ETo calculated from the meteorological stations and EC towers (a), ETo from the DWD data (uncorrected) versus observed ET from the EC tower (b), ETo (corrected) versus observed ET (c), corrected SCOPE_ETo inputs versus observed ET (d), corrected SCOPE_DWD versus observed ET (e), and corrected SCOPE_RS versus observed ET (e). The green dots represent the ROTH site and the blue dots the TUCC site.

Although ET is mainly driven by atmospheric conditions and water availability, the volume of ET depended mostly on the land surface under consideration. The process-based models tested in this study assume a landscape of homogeneous vegetation (e.g. short or high canopy). These models overestimate ET in highly fragmented landscapes with impervious surfaces, as shown in Figure 3. The difference between the two towers emphasises the dependence on the land surface. The ROTH site contains higher vegetation and pervious surface fraction (55 % and 49 %) than the TUCC site (27 % and 28 %). Therefore, the model bias at ROTH is more than twice as low as when the model is applied at TUCC without the correction factor (Table 3). As presented before, the ETo of the two towers is very similar, while the observed ET is twice as low at TUCC.




**Table 3.** Model accuracy for each scenario according to the metrics RMSE, $R^2$ and relative bias for ETo (Penman-Monteith) and SCOPE, with and without the correction factor for urban environments. The highlighted bold values represent the highest precision and lowest bias based on each metric.

| Model approaches | Input scenarios | Correction for urban environments | ROTH | | | TUCC | | |
|---|---|---|---|---|---|---|---|---|
| | | | RMSE | $R^2$ | rBias | RMSE | $R^2$ | rBias |
| ETo | ETo | uncorrected | 0.126 | 0.80 | 1.57 | 0.165 | 0.53 | 3.83 |
| | ETo | corrected | 0.051 | 0.82 | 0.48 | 0.033 | 0.48 | 0.32 |
| SCOPE | ETo | uncorrected | 0.081 | 0.77 | 0.71 | 0.114 | 0.49 | 2.22 |
| | ETo | corrected | 0.033 | 0.78 | -0.007 | 0.024 | 0.45 | -0.12 |
| | DWD | uncorrected | 0.063 | 0.82 | 0.64 | 0.099 | 0.51 | 2.09 |
| | DWD | corrected | 0.026 | 0.83 | 0.05 | 0.021 | 0.47 | -0.16 |
| | DWD+RS | uncorrected | 0.061 | 0.81 | 0.74 | 0.100 | 0.51 | 2.20 |
| | DWD+RS | corrected | 0.026 | 0.82 | -0.001 | 0.021 | 0.47 | -0.13 |

The proposed correction factor for urban environments using GIS information reduces the prediction biases (rBias) and model errors (RMSE) significantly. The corrected ETo prediction, which initially presents a rBias of 1.57 and 3.83 (ROTH and TUCC) is reduced to 0.48 and 0.32, respectively (Table 3). Notice that the RMSE has decreased by a factor of more than two for the corrected predictions, while the $R^2$ value was similar to the original for ROTH. For TUCC, the RMSE was reduced even further, but the $R^2$ is also reduced, which is most likely an artefact as the range of values is smaller after the corrections. Despite the significant improvement using the correction factor, ET prediction based on ETo is still biased, which agrees with other authors that have reported recurrent overestimation from Penman-Monteith models (Allen et al., 2005; Ortega-Farias et al., 2004).

SCOPE model outputs have similar $R^2$ but reduce the relative bias and model error drastically for the corrected predictions compared to ETo predictions. The SCOPE model using the same input variables as the ETo model is much more accurate than the Penman-Monteith model. However, the model accuracy is further improved ($R^2$ of 0.82 and RMSE of 0.026) by the inclusion of other DWD scenario input parameters such as incoming longwave radiation (Rli) and atmospheric pressure (p). The SCOPE models for the RS and DWD scenarios for ROTH present a similar accuracy but lower bias, 0.1 % (RS) against





5 % (DWD). The reduction in bias in the RS scenario can be explained by the inclusion of LAI, which provides a more precise estimation of the vegetation structure in the early season, improving the ET predictions considerably for April.

The selected SCOPE model for TUCC presents an even smaller RMSE (0.021), but also a much smaller $R^2$ and higher bias compared to ROTH. The ET range partially explains these differences in $R^2$ between the two towers, varying from 0 to 0.29 mm at ROTH and from 0 to 0.16 mm at TUCC. The bias values for TUCC can be further reduced if the correction is applied

only during the daytime or if LE soil and LE canopy are corrected by vegetation fraction and impervious fraction separately for the SCOPE model (not shown).

### 3.2 ET seasonality

ET varies greatly across the day and seasons according to changes in meteorological conditions (e.g. temperature, radiation), plant phenology (e.g. LAI, stomatal conductance) and water availability (dry and wet seasons). Figure 4 (c) and (d) shows the

variability in average hourly ET across the months between the two towers (black line). The differences in scale between the two sites are clear, but they present very similar behaviour across time. The predictions using corrected ETo (orange line) overestimate ET from February to October for ROTH and from April to September for TUCC but fit well otherwise. The corrected SCOPE models exhibit the opposite behaviour, being more accurate around the spring-summer and underestimating otherwise.

Observed ET is only higher than predicted ETo in January and December for both sites. The periods when SCOPE models underestimate predictions correspond precisely with the months in which the number of hours of precipitation is higher than the average (Fig. 4a). April was an extremely dry month and all models overestimated ET for both sites, as ET is limited by underground water. A second condition occurs in April, causing a significant increase in vegetation fraction and a decrease in impervious fraction extracted from the footprints at the TUCC site. Atmospheric conditions have led to overall greener

footprints as they were atypically concentrated in a vegetation area (park), reducing the effect of the correction factor without increasing ET values. This phenomenon may occur at TUCC because the tower is located on the top of a building completely sealed with a surrounding wall (Fig. 1) and the effect of dry and wet surfaces are more noticeable there.



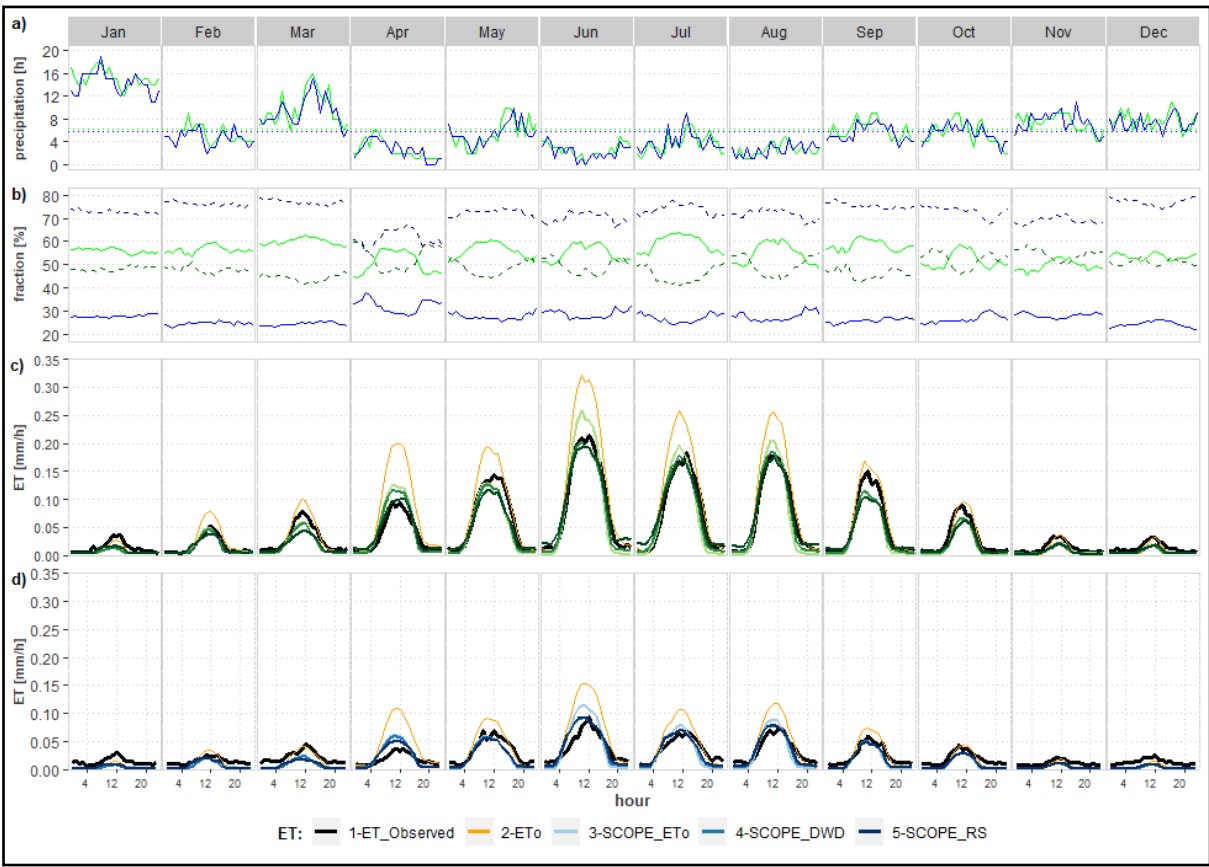

**Figure 4.** Hourly averages per month in 2019 for: (a) precipitation events; (b) percentage of vegetation fraction (solid line) and impervious
fraction (dashed lines); (c) predictions for ROTH site; and (d) predictions for TUCC site. The observed ET (black line) and corrected ETo
(orange line) for both sites. The corrected SCOPE predictions are represented by green lines for ROTH and blues for TUCC site. The light
to dark colours represents SCOPE_ETo, SCOPE_DWD and SCOPE_RS respectively for both sites.

Analysing the time series model accuracy, as expected, the error (mm/h) is not randomly distributed around the zero across
the year. The predictions, in general, are overestimated in summer and underestimated in winter. As both models are
deterministic, temporal autocorrelation in the residuals is not an issue. Yet, their distribution can help to identify in which
conditions the precision and bias in predictions are affecting the overall accuracy. In our case, Figure 5 clearly shows that
model bias is strongly related to the volume of rain over the season.

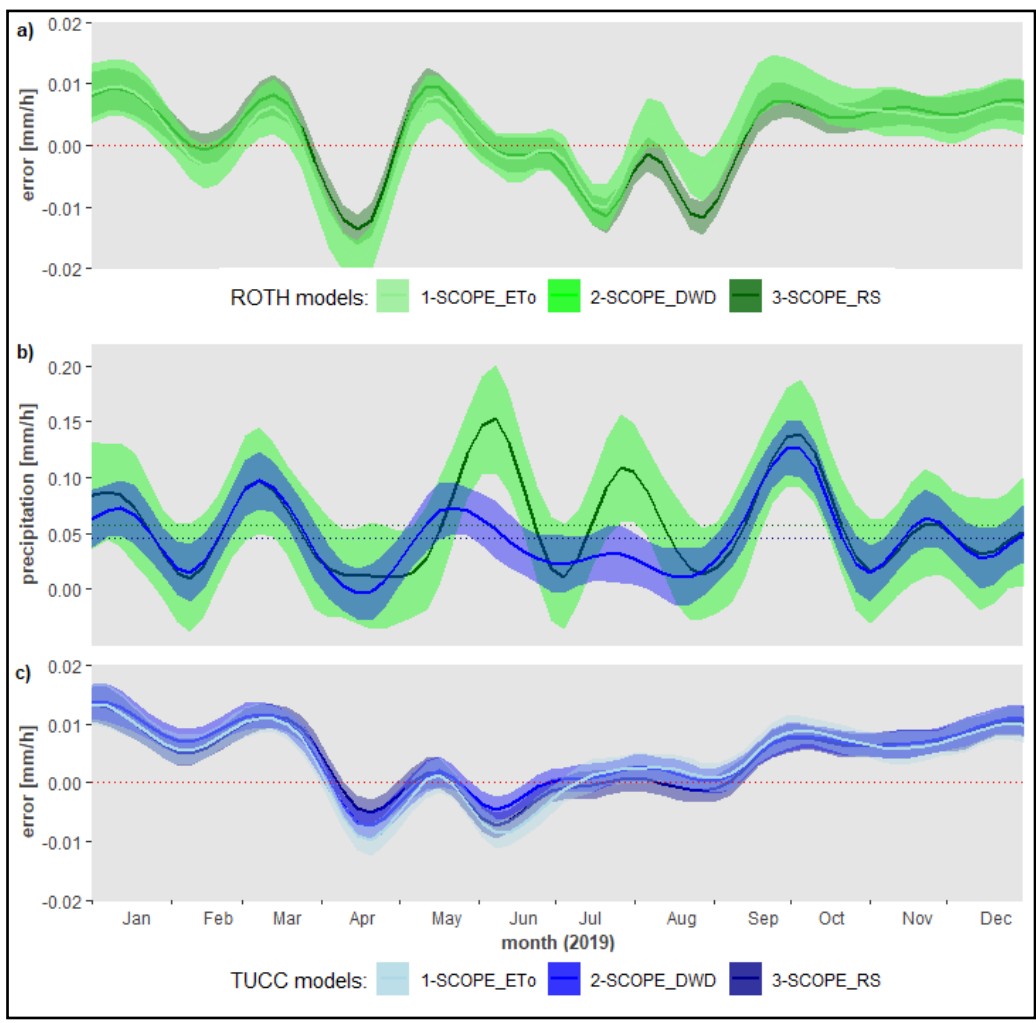

**Figure 5.** Smoothed time series of the volume (mm/h) of precipitation (b) and model error (observed-predicted) for the ROTH site (a) and the TUCC site (c). Green lines represent the ROTH site while blue lines represent the TUCC site. Smoothing function (formula = y~splines::bs(x,20)).

Comparing the SCOPE error curve with the millimetres of rainfall across the year, they present very similar behaviour. When the volume of precipitation is over a certain threshold (around 4mm/h), the ET predictions are underestimated, while below the threshold the model often overestimates ET. The predictions based on ETo are most overestimated during the spring and summer seasons. The year 2019, Germany's third-warmest year since 1881 (German Weather Service - DWD), was extremely dry, which may explain the overestimated values of ET, especially in the most vegetated site (ROTH, Fig. 5b).



### 3.3 Monthly and yearly ET estimations

The data pre-processing and cleaning resulted in around 42 % of missing values in the observed hourly ET because of quality issues or periods of rain (Fig. 2a and e). Even before pre-processing the original ET data from the flux towers, around 10 % of the data were missing. This particular aspect of the eddy covariance method makes it challenging to estimate monthly or yearly ET values. The MDS gap-filling method provides an estimate of 336 mm/year for the ROTH site, which represents 66 % of the observed annual precipitation (Fig 6). This value is very similar to the corrected SCOPE RS model, 328 mm/year or 65 %

of the annual precipitation according to the nearby DWD meteorological station.

  The corrected ETo estimates 477 mm/year (94 %) which seems to be overestimated. At the TUCC site, MDS gap-filling estimates 188 mm/year, which represents nearly half of the annual precipitation volume (47 %), which is much lower than at ROTH. The ETo estimated at TUCC is 236 mm, which represents 59 % of the annual precipitation, while the SCOPE models estimate the lowest values, ranging from 146 to 151 mm/year (36 % to 38 %).

The study "Surface runoff, percolation, total runoff and evaporation from precipitation" presents a long-term mean estimation of evaporation from precipitation for Berlin (Senate Department for Urban Planning and the Environment, 2019). It reports that around 60 % of Berlin's precipitation evaporates and varies from less than 50 mm/year to more than 400 mm/year according to the land surface and water systems available in the region. For the two sites in our study, the evaporation from precipitation was reported as 344 mm/year in the specific location (block) at ROTH and 196 mm/year at the TUCC site,

reducing to 266 mm and 165 mm respectively when considering the average footprint of each tower.

  Compared with the 2013 edition of the Berlin Urban Atlas, the runoff and surface drainage has continued to rise and evaporation has decreased due to the increase in the impervious surface and the expansion of the network for removal of rainwater. The decrease in evaporation due to lack of vegetation and imperviousness can more than double the natural runoff according to the study. It also shows that in winter months the highest rates of percolation occur while in summer the lowest

rates occur. This phenomenon may partially explain why ET is underestimated in the winter season.

  The maximum volume of precipitation for ROTH (i.e. Dahlem station) was observed in June (75 mm) and the minimum in April (6 mm). The maximum value of estimated ET was 95 mm for ETo and 67 mm for SCOPE_RS, also in June, while the minimum was 6 mm and 4 mm respectively in January. The TUCC site (i.e. Tegel station) presents a maximum volume of precipitation in March (62 mm) and the minimum also in April (7.5 mm). The ET estimate reaches the maximum of 49 mm

and 32 mm for ETo and SCOPE_RS models in June and a minimum of 3.2 mm and 2.2 mm in December, respectively.



**Figure 6.** ET by day of the year and hours of the day for the ROTH site. Observed ET after cleaning (a), observed ET gap-filled with MDS (b), Penman-Monteith ETo (c), predicted ET with SCOPE_ETo model (d), predicted ET with SCOPE_DWD model (e), and predicted ET with SCOPE_RS model (f). For TUCC, see Fig. A1 in the Appendix A.

## 4. Discussion

### 4.1 Urban environment and ET

As demonstrated in this study, the use of a correction factor for urban environments provides accurate predictions of ET, similar to the values measured by the eddy covariance method. However, this innovative approach provides a much cheaper and faster method to estimate ET using data from standard meteorological stations in combination with freely available remote sensing data. Data from stations provide very consistent measurements with nearly no missing values, whereas EC data often present a significant amount of gaps. Despite the EC method being the closest attempt to directly measure ET, studies have



reported accuracy varying from 5 % to 20 % (Foken, 2008; Liang and Wang, 2020), which may be even higher in urban environments as the lack of energy balance closure is more pronounced.

We also showed that similar atmospheric conditions can produce very distinctive ET values as the process is highly dependent on the land surface of the site under consideration. As a combination of evaporation from soil, intercepted precipitation and plant transpiration, the volume of water produced from ET varies greatly according to the vegetation cover and imperviousness. Classical process-based models using the Penman-Monteith equation focus mostly on the atmospheric interfaces, lacking representation of the land surface. The SCOPE model, as part of the SVAT model family, allows for a comprehensive

parameterisation of soil and vegetation properties (van der Tol et al., 2009). However, these models assume a landscape composed of homogeneous vegetation without anthropogenic elements. Urban environments are highly fragmented surfaces (vertically and horizontally), containing heterogeneous vegetation types and heights combined with varying levels of imperviousness (Nouri et al., 2013).

The effect of surface heterogeneity in the horizontal direction, typical in an urban environment, is not addressed by (1D)

models such as SCOPE or Penman-Monteith-based models. Vertical heterogeneity in the canopy can be specified using mSCOPE (Yang et al., 2017). Still, in urban environments, buildings are one of the primary sources of surface roughness length and displacement height, altering the wind speed and direction locally (Grimmond and Oke, 1999). Detailed land cover information derived from high-resolution remote sensing products increasingly available in cities, combined with the capacity of SCOPE to account for surface-atmosphere interactions, proved to be a feasible approach to predicting ET in urban

environments.

## 4.2 ET time series and model validation

Terrestrial ET is a very complex variable to be measured as it is a combination of distinctive processes such as evaporation from soil moisture and precipitation intercepted by canopies, impervious surfaces and plant transpiration. Miralles et al. (2020) advocates that despite all the processes being essentially evaporation, interception loss, soil evaporation and plant transpiration

should be assessed and modelled separately. However, the complexity of ET requires simplifications of many assumptions without an obvious solution.

The most common instruments to provide a time series of ET measurements are eddy flux towers, lysimeters, and sap flow (Nouri et al., 2013). These instruments are focused mostly on one of the evaporation processes, which may require combining them to entirely understand ET. Eddy covariance, which was used in this study, is one of the most suitable approaches for

measuring ET, especially in urban areas (Foltýnová et al., 2020; Nouri et al., 2013). Measuring latent heat flux from the atmosphere, LE measurements can easily be converted to ET and comprise evaporation from soil, transpiration and interception loss together.

Nevertheless, there are some drawbacks to EC measurements, such as (1) LE is a measurement of energy to transform water to vapour (i.e. ET) but also the other way around (condensation), which produces negative ET values; (2) The source area

representing ET measurements varies continuously in size and shape, which is an issue in heterogeneous urban environments





(Kljun et al., 2002; Kotthaus and Grimmond, 2014; Schmid and Oke, 1990); (3) During rain and after a certain subsequent period, EC measurements are not reliable, presenting unrealistically high values of ET (Kotthaus and Grimmond, 2014; Ward et al., 2013); and (4) Anthropogenic vapour emissions such as car exhaust or building heating are accounted as ET as well (Karsisto et al., 2016; Kotthaus and Grimmond, 2012; Nordbo et al., 2012; Ward et al., 2013).

However, sap flow and lysimeter measurements present even more severe limitations than flux towers for urban environments (Litvak et al., 2017; Nouri et al., 2013). Lysimeters can be a proxy of the ET derived from soil evaporation and the water uptake by plants (converted to biomass or transpired). Furthermore, lysimeters are generally not well-suited to the heterogeneous vegetation and complex impervious surfaces common in urban areas as they are a point-based measurement and are installed underground (Nouri et al., 2013). Sap flow is related only to transpiration as it measures the water uptake by

an individual plant. This method is feasible for trees and is species-specific, which limits its applicability for urban vegetation, which is composed of a mixture of grassland, shrubs and various tree species (Nouri et al., 2013).

The ET concept also raises open questions, such as whether ET during rain or at nighttime should be considered zero and to what extent all mentioned evaporation processes exhibit similar seasonality. Soil evaporation and plant transpiration have similar drivers and their daily values are strongly correlated in energy-limited regions, but the same does not apply to

interception loss (Martens et al., 2017; Miralles et al., 2020). For instance, despite the ROTH site presenting a significantly higher overall ET and vegetation fraction, the average ET at night at TUCC is higher than at ROTH for all seasons. Based on Figs. 4 and 5, the interception loss could explain this observation, as the urban canopy may intercept more precipitation than the vegetation canopy. Precipitation intercepted in forests at continental scales is reported to be similar in magnitude as soil evaporation, reaching 10–20 % of the total latent heat flux over land (Miralles et al., 2020; Wei et al., 2017). In addition,

vaporization of intercepted precipitation often exceeds daytime rates of transpiration even at nighttime. Furthermore, as interception loss is mostly driven by rainfall and land surface properties, it is less constrained by net radiation and therefore not accounted for by SCOPE and Penman-Monteith models (Webb et al., 1980).

Transferring these findings to urban environments, with the presence of anthropogenic heat fluxes such as building heating systems and transportation, interception loss may play an even more substantial role than in natural environments, especially

in winter and at nighttime. According to Ramamurthy and Bou-Zeid (2014), wet impervious surfaces evaporate at higher rates than wet vegetation, as they often store more heat. They conclude that evaporation from wet impervious surfaces such as concrete pavements, asphalt and building rooftops accounted for around 18 % of the LE and may last up to ten days, with the highest evaporation rates occurring 48 hours after a precipitation event. In our study, we have excluded data up to 4 hours after precipitation events from the validation dataset to assess model accuracy.

However, as interception loss and precipitation are not part of the SCOPE model, the monthly and annual ET values may be underestimated. This underestimation is likely to be particularly pronounced in the highest built-up areas, as reflected by the relative bias at the TUCC site (-0.13). Furthermore, in winter time, when dew point temperatures are most common, condensation generates wet surfaces at night (cars, windows, roads, metal roofs), which evaporate again during the day, increasing the ET measured by the EC method without being uptake by the soil or trees, similarly to interception loss.





The discrepancy between the concept of ET and its direct measurements with the available approaches makes model validation challenging. Some of the model bias could be attributed to the flux tower measurements. For instance, the underestimation of the model predictions in the winter and periods with higher precipitation could be an artefact of the bias from observed ET. Ward et al. (2013) also indicates that directly following rainfall, LE measured by the EC method presents significantly higher values than modelled LE. LE measurements from EC towers are reported as slightly underestimated due to the lack of closure in energy balance caused by low turbulence (Kracher et al., 2009). Therefore, it is a common practice (not applied here) to

correct LE values using the residual or Bowen ratio methods (Liang and Wang, 2020).

A combination of sap flow with flux tower measurements could facilitate a better understanding of under what circumstances atmospheric conditions (in particular radiation and precipitation) ET measurements based on eddy covariance agree with the water uptake by trees. The use of gas chamber exchange to better understand the daily and seasonal variation of model inputs

related to photosynthesis such as stomatal conductance (m), respiration and maximum carboxylation capacity (vcmax) could also further improve the model validation.

Remote sensing data can help to overcome the difficulties in obtaining SCOPE model inputs such as LAI, chlorophyll (Chl) and stomatal conductance in different seasons. In our study, the inclusion of the LAI Copernicus product reduced the bias in ET predictions in the early spring. Incorporating LAI was particularly beneficial in April, as using the default constant of LAI

overestimates ET: in April, the temperature starts to increase but the canopy foliage is still incomplete. The inclusion of canopy height in combination with LAI, slightly improved the model accuracy in general. Model accuracy was also significantly improved when the option to correct the parameter vcmax by the hourly temperature was selected. Our results show that the seasonality of photosynthetic parameters is highly important for ET estimates using SCOPE. Other studies have derived vcmax from chlorophyll content retrieval by remote sensing data to include in the SCOPE model (Houborg et al., 2013; Wolanin et

al., 2019).

The temporal variability of ET makes it challenging to align all of the essential SCOPE parameters in space and time with remote sensing data (Pacheco-Labrador et al., 2020). However, model inputs which are not significantly affected by daily variability such as LAI, leaf pigments, water content, and dry matter content can be reasonably supplied by remote sensing data (Raj et al., 2020). As shown in this study, interpolating the RS derived input parameter (i.e. LAI) from 10 days resolution

to hourly have contributed to the model accuracy. Also SCOPE automatically calculates the effect of solar angles on the fraction of sunlit and shaded leaves, which reduces the impact of the time lag difference between the spectral data and ET observations across the year driven by the fluctuation in sun zenith angle.

### 4.3 Model generalisation and upscaling

The advantage of a process-based model (i.e. fully deterministic) over an empirical model is that training is not required, which

increases the chances of generalising the model to other locations. Thus, SCOPE could be used to spatially upscale ET to the city level. We demonstrate here that one meteorological station is enough to provide input variables to characterise the atmospheric conditions for different locations in a large city such as Berlin. For instance, the incoming solar radiation inputs



(shortwave and longwave) used in this study were provided by a DWD station located in another town (Potsdam) more than
20 km distance from both sites. The spatial mismatch is not so important for atmospheric conditions as having a high temporal
resolution (e.g. hourly). In contrast, for remote sensing and GIS data, avoiding spatial mismatch and providing an adequate
spatial resolution of the land surface is critical.

The difficulty of scaling ET measured by eddy covariance (up or down) is that the observations are not point-based, nor
corresponding to a regular grid or plot area. The source area of ET varies in size and shape according to the wind speed and
direction interacting with the surface roughness and displacement height at a specific snapshot of time. Therefore, footprints
for every timestamp are required to relate the ET values to a land surface, making it challenging to scale ET values to a regular
grid without flux tower measurements of wind profile. In this study, the relation of ET with vegetation fraction and impervious
fraction extracted from the footprints shows a moderate correlation (0.35 and -0.44) for the ROTH site but no significant
correlation for the TUCC site. In summer, the percentage of vegetation fraction increases during the day up to noon, while the
impervious fraction presents the opposite behaviour in the ROTH site (Fig. 4b), which may partially explain the better
correlation. Based on the results at the TUCC site and impossibility to validate the footprints in a highly heterogeneous surface,
the use of a buffer or pixel window may achieve similar results. Alternatively, a simpler footprint derived from wind speed
and direction freely available from meteorological stations could be applied without losing significant accuracy (see Quanz
(2018)).

Time series of spectral information derived from satellites could also be used to retrieve plant traits in the inverse modelling
approach for parameterisation of biochemical and biophysical inputs in SCOPE (Raj et al., 2020). However, plant trait
measurements would then be necessary, which would necessitate training the model locally (semi-deterministic) and therefore
reduce the capacity to generalise and upscale the model. However, better parameterisation of vegetation properties may
increase the model accuracy compared with the default parameter values used in this study. The network of DWD stations
could be used to create spatiotemporal raster layers with the primary inputs of atmospheric conditions required to model ET
using the same grid resolution as the land surface data. The combination of high temporal resolution raster data of atmospheric
conditions and land cover surface data with high spatial resolution can make it feasible to produce accurate ET maps for entire
metropolitan regions.

Hydrological models can contribute detailed information on the rainfall–runoff relationship and water storage processes,
including infiltration, percolation, depression storage, and groundwater flow, among others. Although they often include
evaporation processes (soil evaporation, plant transpiration and interception loss) separately, they are designed to understand
river basins or plan urban drainage systems. These models often estimate potential or reference ET and then convert it into
actual ET as a function of Soil Moisture (Zhao et al., 2013). The potential ET estimation often uses Penman-Monteith
equations, but different models use approaches based on the energy-based, temperature-based and mass transfer-based
methods. Therefore, their outputs are rarely comparable (Zhao et al., 2013).
Distributed hydrological models are more suitable for heterogeneous urban environments as they can model an area based on
a grid, but they are very complex, demanding significant amounts of data inputs and time to apply them correctly. With extreme





precipitation events and heat island effects tending to occur more spatially localized and less temporally predictable, hydrological models may be over-complex to model urban ET at high spatiotemporal resolution. Some distributed hydrological models provide only yearly or monthly ET values, while just a few give sub-daily estimates.

The hydrological model for Berlin (ABIMO 3.2) compared to our results in this study provides detailed spatial resolution, comprising very comprehensive information about the city, including land use and land cover, drainage systems, groundwater, and green roofs. However, the yearly temporal resolution (not updated every year) is much broader than required to mitigate the UHI effect and extreme events. Our study arrives at similar annual values of ET using a much simpler approach, while providing accurate ET estimates at an hourly scale. Although annual ET predictions are difficult to validate with EC

observations, as the accuracy of the hourly estimations are unreliable at certain conditions such as non-steady-state or absence of well-developed turbulence (i.e. quality flag 2), and during precipitation events or in the following hours.

**4.4 Applications and recommendations for the future**

The accuracy (i.e. bias) for the predictions in spring/summer and autumn/winter performed differently, with the models in wintertime clearly underestimating ET. Modelling SCOPE separately for each season may improve the accuracy as

aerodynamic, photosynthetic, soil, and canopy constants could be better specified for these periods. Given that most of the applications to model ET are constrained to the growing season, constants and default parameters are likely to be optimised for these conditions (Ward et al., 2016). SCOPE has more than sixty model inputs, allowing for greater model customisation to the local environment than presented in this study. However, the objective of this study was to demonstrate the potential of SCOPE in urban environments when using our proposed correction factor, rather than to provide a final model for Berlin.

Thus, most of the parameters were kept constant using default model values.

Tuning or measuring most of the input parameters to match the reality of the specific urban environment under consideration can further improve the model accuracy. Process-based models (SCOPE or hydrological) provide the opportunity to perform sensitivity analysis and simulate scenarios to understand the impact of underlying drivers of ET on the accuracy. The SCOPE LE output can be separated into soil and canopy components, which can be used to further improve the correction factor as

transpiration and evaporation are likely to differ across seasons and land surfaces. By separating soil LE, different kinds of impervious surfaces could be better investigated. These improvements could facilitate future applications, producing maps and ET indicators and publicising them systematically.

Applications providing accurate ET maps could support a variety of initiatives, from controlling irrigation for managing green spaces in cities to planning more sustainable urban environments. Such maps could also assist local governments in mitigating

UHI effects, reducing health risks during extreme summer temperatures. Smart and green city initiatives could utilize dynamic ET maps to monitor the impact of climate change and identify solutions to improve the quality of life in cities worldwide. A better understanding and management of the water cycle (green, blue and grey) will be vital for human well-being in the near future.



## 5. Conclusion

This study has proposed a novel approach to estimate hourly evapotranspiration (ET) in urban environments using the SCOPE model. Although SCOPE has successfully been applied to predict ET in previous studies, this is the first time that SCOPE has been applied in an urban environment. Most process-based model approaches to estimating ET, including SCOPE, are designed for homogeneous vegetated landscapes, resulting in overestimation of ET in urban areas. We developed a correction factor for urban environments using land surface derived from remote sensing and GIS data, which proved to be efficient in reducing

model bias and improving global accuracy. This approach, which combines high temporal resolution data of atmospheric conditions from meteorological stations and high spatial resolution data of land surface derived from remote sensing, paves the way to creating ET maps systematically for urban and metropolitan regions. Such maps are highly relevant to urban planning and climate change mitigation aiming to reduce the urban heat island effect and to sustainably manage water resources in cities.

**Acknowledgements.** This work was supported by the German Research Foundation (DFG) within the Research Training Group 'Urban Water Interfaces' (GRK 2032-2). The German Federal Ministry of Education and Research (BMBF) funded instrumentation of the Urban Climate Observatory (UCO) Berlin under grant 01LP1602 within the framework of Research for Sustainable Development (FONA; 635 www.fona.de). The authors would like to thank the DWD, the Chair of Climatology at the Technische Universität Berlin, the European Commission, and the Berlin Senate Department for Urban Development and

Housing for providing data used in this paper. We would additionally like to thank Fred Meier for pre-processing and providing the eddy covariance data and Justus Quanz for providing R code to optimise footprint modelling.



## Appendix A

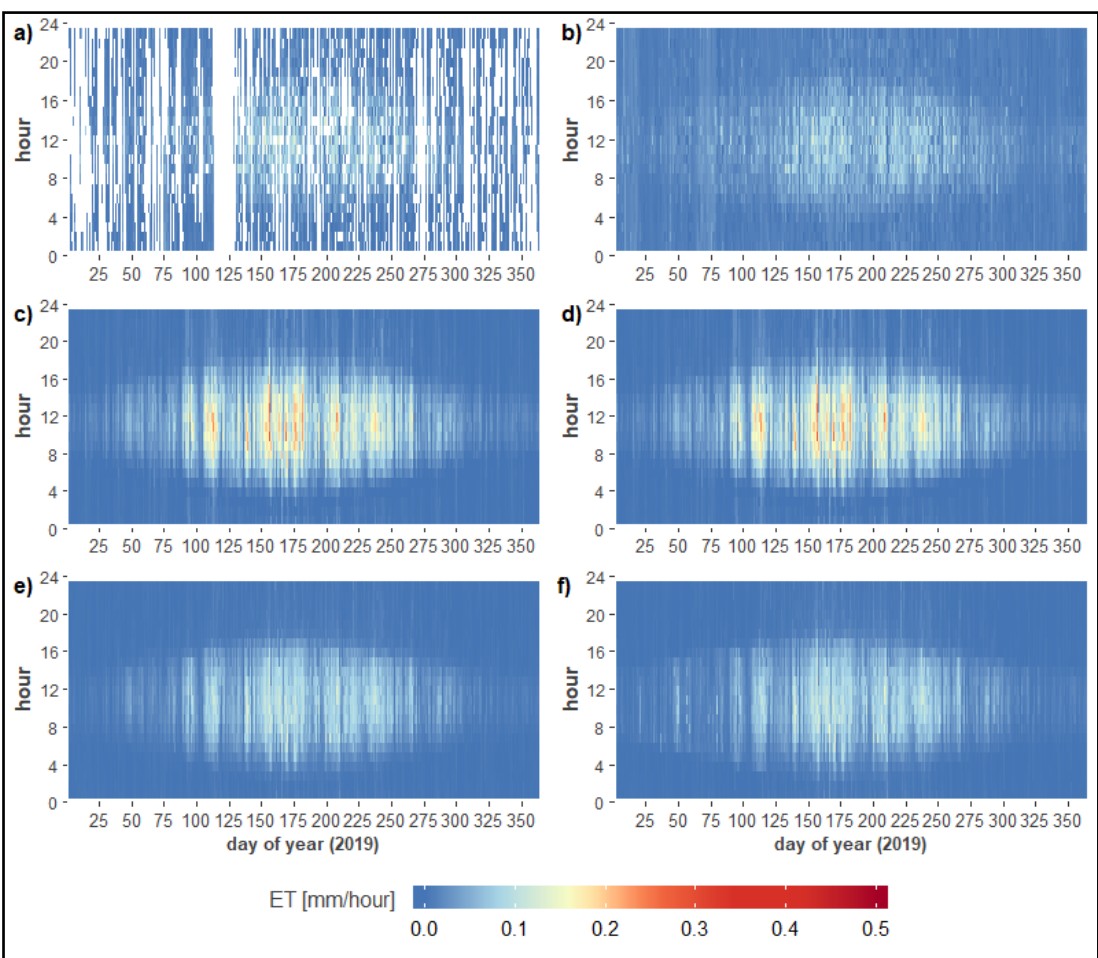


**Figure A1.** ET by day of the year and hours of the day for the TUCC site. Observed ET after cleaning (a), observed ET gap-filled with MDS (b), Penman-Monteith ETo (c), predicted ET with SCOPE_ETo model (d), predicted ET with SCOPE_DWD model (e), and predicted ET with SCOPE_RS model (f).

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
