# Peer review of "Modelling hourly evapotranspiration in urban environments with"

_Hydrology and Earth System Sciences, 2021_

## Referee Comment (RC1)

**Review of "Modelling hourly evapotranspiration in urban environments with SCOPE using open remote sensing and meteorological data"**

*Summary*

In the manuscript, the authors present a methodology to improve urban evapotranspiration modelling by adapting the SCOPE model for urban environments and adding remotely sensed data. They successively supply different levels of information to the SCOPE model and compare its performance. They conclude the model is capable to predict hourly values for urban evapotranspiration based on the adapted model supplied with open remote sensing and meteorological data.

*General comments*

The topic of ET in urban areas is, as explained by the authors, a very relevant and interesting one that differs from ET in more homogeneous areas. Predicting urban ET is difficult, especially on the hourly scale, but the authors manage to achieve satisfying results with their adaptation of SCOPE. Remotely sensed data is not widely applied in the modelling of ET in urban environments, while it is shown here to have potential to improve the quality of predictions. I am concerned about the use of the SCOPE model, since there are models that are readily adapted to urban areas, but the authors decide not to use these for unmentioned reasons. In addition, the quality of the manuscript itself is not of the level it should be.

In my opinion, this paper presents very relevant findings, but the manuscript is not clear and readable enough. In addition, a model is chosen that is not the most appropriate. Therefore, the comments below should be addressed to improve the quality of the text and thoroughness of the research before it could be published.

**Structure of the text**

The introduction is clear in its content and covers most of the relevant literature. It frames the research nicely and puts it in the right perspective. However, there is room for improvement regarding the storyline, which is now mostly missing. Paragraphs miss a clear subject, their connections are not described and their order is not always logical (introducing EC and only later the less suitable alternatives). L36-44 is good example, where I am not sure what the message is: ET is both mass and energy; the composition of terrestrial ET or the different observational techniques. Some of these messages also come back in the next paragraph. In this case, I would reorder the paragraphs to separate the explanation of the ET process and the observations. This is not the only paragraph where this issue occurs. The same can be seen in the abstract, which is a summation of facts that are not actively linked. This leaves it up to the reader to fill in the blanks.

The structure of the manuscript stays a limiting factor in its understanding beyond the introduction. Apart from links between paragraphs, short phrases explaining what will be said in a section have the potential to improve the manuscript. I will comment on the textual side in more detail, once the structure has been improved.

**Missing literature**

An elaborate overview of models is given (hydrological, surface energy models, L70-87). It feels like the models designed for the cities are overlooked, while they would be appropriate models. Grimmond et al., (2011) evaluated this type of models concluding ET is the most poorly modelled flux

making the analysis of the authors even more interesting. The authors state (L94): "Most ET modelling approaches assume a landscape of homogeneous vegetation without anthropogenic elements." By which they omit a significant part of the urban climate research field. This is also missing in the discussion, where SVAT and hydrological models are discussed, but no urban climate models.

**Choice for SCOPE**

This brings me to a related point and my main concern, which is the missing argumentation for the choice for SCOPE. The inclusion of vegetation and soil in the model is interesting, but it is not developed for the city. Other models UT&C (Meili et al., 2020) and SUEWS (Järvi et al., 2011) also include vegetation and soil, but are designed for urban areas already. Given the authors argue that the model is not suitable for urban areas (L229-230), why is SCOPE adapted for urban areas instead of using remote sensed input for a readily adapted model? SCOPE is not the most appropriate model for the study at hand, and the author should consider redoing the analysis with a model designed for urban areas.

**Clarity of methodology**

While the actions taken reasonably well described, the methodology lacks explanation of the design choices. This raises questions for example regarding why data was left out and altered in a certain way. The effect of the choices should also be discussed. In the minor comments below, I indicate examples of choices for which the reasons are not clear. Also, for some parts references or explanations are missing, which is also indicated below.

**Correction factor**

Defining the correction factor as the vegetation fraction creates an assumption that is not mentioned in the manuscript. By multiplying the predicted ET by the vegetation fraction it is assumed that this fraction produces all relevant ET. At the same time, literature shows impervious surfaces make a significant contribution to ET (Ramamurthy et al., 2014; Wouters et al., 2015). Therefore, this limits the conditions for which the presented model can be applied. In case of completely impervious area, the correction would lead to a permanently zero ET, which is not realistic.

In the discussion, the authors give a very complete framing to the results. The discussion also includes a lot of repetition and remarks that do not seem to have a purpose. Therefore, I think it can be shortened considerably. Implications of drawbacks and assumptions that are mentioned for the results are often not mentioned. The conclusion is very to the point, but should be more specific regarding amongst others the performance of the model.

The figures properly support the text and show the results clearly. I especially liked Figure 6. Unfortunately, the image quality of the figures is poor. For some, the images are grainy and individual pixels can be seen. They are also not easy to read due to missing legends and titles for panels. The captions are very long with a lot of information on how to read the figures. Legends and subtitles could help to transfer some of this information to the figures themselves. In the minor comments I included remarks per figure.

*Minor/technical comments*

Apart from these larger concerns regarding the methodology and reflection, I have some smaller points that could help improve the manuscript. Also, I wrote down a list of textual improvements,

which is not meant to be an exhaustive overview. Since I think the writing needs major adjustments, I have not corrected the language in detail. The points are indicated with a line number.

L11-13: It sounds contradictory that most models assume a homogenous vegetated landscape, but at the same time they lack input parameters to describe the land surface.

L22: It is unclear to me what the respectively refers to, is that the different sites?

L24-25: I would like to see a more concrete suggestion for an application, instead of a very general statement that fits under almost all urban climate research. What does this research specifically add?

L30: I think the adverse health effects of the UHI are better described in other papers than Vulova et al., 2020. Either remove or replace this citation, since the other two are appropriate.

L34: In my opinion, it is not substantiated here why ET us so important in urban areas, although I do fully agree with the statement itself.

L63: A practical solution to what problem?

L101: Why is the choice here to use evaporation, while in the whole manuscript evapotranspiration is used?

L104-105: This sentence is not correct English.

L115-118: The use of comma's would improve the readability of long sentences like this (or consider breaking them up).

L123: Omit "(mm)".

Figure 1: The color scale for the impervious fraction is hard to read due to the use of two colors that do not have a different meaning. It would also not be readable in black and white.

The buffers are drawn at 1500 m, but in literature 500 m is used more often (e.g. Coutts et al., 2007 and Hong et al., 2020).

It could be useful to use letters for the different panels to ease referring to parts of the figure.

L136-139: The second sentence repeats much of what is said in the first one (e.g. a sonic anemometer measures orthogonal wind components).

L140: I would say this is part of a site description, not the data.

L146: The authors state: "Negative values (condensation) were set to zero." I think this affects the results, while it is not based on anything physical. What are the effects on the results? Has an analysis without setting all these values to zero been performed.

L146-148: The used prepositions seem odd here.

L151-153: The choice to leave out the 4 hours directly after precipitation will cause a problem, since at this time ET is especially high in cities (Ramamurthy et al., 2014). What is the influence on the conclusions?

L155-157: What methodology was used to calculate the footprints?

L173: This is not the first point the content of Table 1 is discussed.

L179: Why is linear interpolation performed if the differences are minor and irrelevant to the study?

Table 1: This table summarizes a lot of information very nicely. It also feels like a lot of repetition from the text, which provides the opportunity to shorten the methodology.

L188-190: Repetition of L155-157.

L196-197: This method is not clear to me, what pixels are used and which are not?

L198: What was the original resolution of this dataset? And is this one map for the entire year?

Figure 2: Please include a legend indicating the meaning of colors and line style (it is now only in the caption).

L221-223: Given that this is the definition of benchmarking (Best et al., 2015), it would be good to use that term.

L227: This sentence is not correct English.

L229: In my view, the model will not provide but require a lot of input, which in turn can be positive, but also has the potential to limit the usability of a model.

L233: Replace "group" with "groups".

L237: What are these other parameters at the end of the line? The same as the ones mentioned earlier?

Table 2: The unit of air pressure is stated as ppm, but this seems odd to me since this is not the standard unit.

L255: The direction of bias would be good to include.

L257-259: Is the vegetation fraction the correction factor? It seems logical, but it is not stated explicitly.

L259-260: This is stated more elaborately in L263-269.

L263: All available ET values observed before or after the filtering described earlier?

L263-265:

L266: What is the definition of rBias?

L272-273: Repetition of L75-76.

L274-275: The reference is to only one panel of the figure (3a), while results for any combination are stated. What are all the possible combinations and are these all shown?

L277-278: A very interesting result that could be highlighted more given its wider potential for application.

Figure 3: Panel a has a y-label stating DWD data and the others simply state DWD. Are these the closest DWD stations to the flux towers or were multiple combinations tested? On top of that, panel b-c show ET_0, but in d-f this is replaced by predicted ET. $ET_0$ is in this manuscript also treated as a prediction. A minor detail is the inconsistency of the capitals in the x-labels.

In addition, panel a does not show the observed ET on the x-axis. Panels b-f tell a story together (evolving precision using different approaches), but panel a is different and as shown now feels as if it should be a separate figure.

This figure may benefit from titles for the subpanels to in one glance see the differences between the plots.

L285: This sentence contradicts itself. If atmospheric conditions are the main driver, why is it mostly dependent on the land surface?

L286-287: Repetition.

L288: Please add panels to the figure reference, since some are corrected for this assumption.

Table 3: The caption states that the highest precision and lowest bias are highlighted in bold, but this is not shown in the table.

The names of the scenarios may cause confusion, since $ET_0$ is both a model approach and an input scenario. For example, in L303, it is bit clear without checking the values in the table what is the corrected $ET_0$ prediction.

L317: What is meant with the word "selected" in front of SCOPE?

L320: This suggest SCOPE produces output for specific sources of ET, which would be very interesting. Is this correct?

L324: Panels 4c and d are referenced before a and b.

L330: The observed ET may be influenced by an anthropogenic moisture flux. How relevant is this at the study sites?

L332-333: Is ET prediction in SCOPE linked to the water availability?

Figure 4: A very insightful figure, but for panel a and b a legend is missing.

Figure 5: In order to clarify the link between the error and the precipitation, showing them in the same graph (reducing the panel count to 2) makes comparison easier.

What is the meaning of the dotted lines in panel b?

L360-363: Repetition.

L370: In order to value the comparison of the study at hand and the on done by the Senate Department for Urban Planning and the Environment, some background of that study should be discussed.

L375: I miss a comparison with the results of the work from the authors.

L378-379: To what study is referred here?

L406-408: Repetition.

L409-415: The message of this paragraph is not clear to me.

L417-427: Repetition of the introduction.

L435-441: Since none of these observation methods is used, why is this relevant for the study at hand?

L453-455: I agree with the authors that the interception loss/evaporation will be important in urban areas, but the link to anthropogenic heat fluxes should be explained.

L458-462: To my understanding, removing these 4 hours would decrease the value of the modelling approach for water balance estimation, as the interception evapotranspiration is not (completely) take into account. As is also stated in the manuscript, but I do not understand why as a consequence these 4 hours are excluded from analysis.

L462: There is always a dew point temperature.

L462-464: I do not see why interception loss is comparable to water that is condensed and evaporates again or what the relevance is of that it is never taken up by the soil or a tree. Additionally, I doubt whether the water that evaporates from the mentioned wet surfaces has been detected by the EC systems, since they are installed high above the surface.

L466-467: To what bias is referred here and what is the source?

L468-469: What are the implications of this difference? Are the models wrong or the observations or is something else going on?

L470-471: If this correction is common practice, why is it not applied?

L473-492: There are a considerable number of interesting and noteworthy statements in here, but I miss the connections.

L494-501: In my opinion, this is a very important outcome of the study. It paves the way for the prediction of ET for cities without highly specialized observation equipment, such as EC. It deserves a spot light in the conclusion.

L506-508: I was surprised by the clear daily cycle in the composition of the footprint. I assume this is related to the daily cycle of the wind direction and strength, but a short background on this cycle would help to interpret the results.

L511-512: Has this been tested for the approach in this study?

L529: What is not comparable in what terms?

L542: I value the inclusion of a future perspective. However, I think in the previous subsections there are already parts of this perspective. It would help to keep them together. Also, part of the section itself are not a future perspective, but an evaluation of this study and how it was adapted to its goals.

L569: Please further specify "land surface".

L544-545: The verbs switch time in this sentence, so it is not clear whether this is a general statement or one on this study.

Figure A1: This figure shows the data for April are missing to a great extent, but more nighttime observations seem to be there. Could this partly explain the poor performance during this month?

L732-733: This link does not work.

*References*

Best, M. J., Abramowitz, G., Johnson, H. R., Pitman, A. J., Balsamo, G., Boone, A., Cuntz, M., Decharme, B., Dirmeyer, P. A., Dong, J., Ek, M., Guo, Z., Haverd, V., van Den Hurk, B. J. J., Nearing, G. S., Pak, B., Peters-Lidard, C., Santanello, J. A., Stevens, L., & Vuichard, N. (2015). The

plumbing of land surface models: Benchmarking model performance. *Journal of Hydrometeorology*, *16*(3), 1425–1442. https://doi.org/10.1175/JHM-D-14-0158.1

Coutts, A. M., Beringer, J., & Tapper, N. J. (2007). Characteristics influencing the variability of urban CO2 fluxes in Melbourne, Australia. *Atmospheric Environment*, *41*(1), 51–62. https://doi.org/10.1016/j.atmosenv.2006.08.030

Grimmond, C. S. B., Blackett, M., Best, M. J., Baik, J. J., Belcher, S. E., Beringer, J., Bohnenstengel, S. I., Calmet, I., Chen, F., Coutts, A., & others. (2011). Initial results from Phase 2 of the international urban energy balance model comparison. *International Journal of Climatology*, *31*(2), 244–272.

Hong, J.-W., Lee, S.-D., Lee, K., & Hong, J. (2020). Seasonal variations in the surface energy and CO2 flux over a high-rise, high-population, residential urban area in the East Asian monsoon region. *International Journal of Climatology*.

Järvi, L., Grimmond, C. S. B., & Christen, A. (2011). The Surface Urban Energy and Water Balance Scheme (SUEWS): Evaluation in Los Angeles and Vancouver. *Journal of Hydrology*, *411*(3–4), 219–237. https://doi.org/10.1016/j.jhydrol.2011.10.001

Meili, N., Manoli, G., Burlando, P., Bou-Zeid, E., Chow, W. T. L. L., Coutts, A. M., Daly, E., Nice, K. A., Roth, M., Tapper, N. J., Velasco, E., Vivoni, E. R., Fatichi, S., & others. (2020). An urban ecohydrological model to quantify the effect of vegetation on urban climate and hydrology (UT&C v1.0). *Geoscientific Model Development*, *13*(1), 335–362. https://doi.org/10.5194/gmd-13-335-2020

Ramamurthy, P., Bou-Zeid, E., Cha, Y., Park, S. S., Kim, K., Byeon, M., & Stow, C. A. (2014). Contribution of impervious surfaces to urban evaporation. *Water Resources Research*, *50*(4), 2889–2902. https://doi.org/10.1002/2013WR014979.Reply

Wouters, H., Demuzere, M., De Ridder, K., & van Lipzig, N. P. M. (2015). The impact of impervious water-storage parametrization on urban climate modelling. *Urban Climate*, *11*, 24–50.

---

## Author Response (AR1)

We appreciate the time and effort made by the editor and the reviewers to critically evaluate our manuscript. Based on the feedback from the reviewers, we carefully adjusted the manuscript where possible to address the reviewer's comments. Below we present a point-by-point summary of all edits made to address these suggestions.

The major changes were in the introduction, revision of the figure labels and colour, and the rewriting of the Discussion. We provided some extra information in the methodology, also we replaced a few terms and fixed some grammar issues following the reviewer's suggestions.

The reviewer's comments are in orange colour, while in black are our answerers and in red the alterations in the text that can be found in track changes file.

**Reviewer #1**

**Summary**

In the manuscript, the authors present a methodology to improve urban evapotranspiration modelling by adapting the SCOPE model for urban environments and adding remotely sensed data. They successively supply different levels of information to the SCOPE model and compare its performance. They conclude the model is capable to predict hourly values for urban evapotranspiration based on the adapted model supplied with open remote sensing and meteorological data.

**General comments**

The topic of ET in urban areas is, as explained by the authors, a very relevant and interesting one that differs from ET in more homogeneous areas. Predicting urban ET is difficult, especially on the hourly scale, but the authors manage to achieve satisfying results with their adaptation of SCOPE. Remotely sensed data is not widely applied in the modelling of ET in urban environments, while it is shown here to have potential to improve the quality of predictions. I am concerned about the use of the SCOPE model, since there are models that are readily adapted to urban areas, but the authors decide not to use these for unmentioned reasons. In addition, the quality of the manuscript itself is not of the level it should be.

In my opinion, this paper presents very relevant findings, but the manuscript is not clear and readable enough. In addition, a model is chosen that is not the most appropriate. Therefore, the comments below should be addressed to improve the quality of the text and thoroughness of the research before it could be published.

We are grateful for your very detailed review and valuable suggestions for our study. We have addressed most of your concerns, revising the manuscript accordingly with your suggestions or explaining the reasons otherwise. The manuscript was proofread by a native English speaker.

The main concern about the choice of the SCOPE model and some of our assumptions in the study were addressed in the reply to the reviewer, and as suggested by the editor, provided a major revision based on the reply posted in the discussion phase. This study provides a solution that combines the temporal dynamic of ET in a vegetated environment with the spatially fragmented land cover in urban environments, a less computationally expensive but plausible ET product suitable for most cities in the world. The prediction accuracy (precision and bias) is compatible with the state-of-the-art in urban models but is potentially more transferable as the approach uses open data.

**Structure of the text**

The introduction is clear in its content and covers most of the relevant literature. It frames the research nicely and puts it in the right perspective. However, there is room for improvement regarding the storyline, which is now mostly missing. Paragraphs miss a clear subject, their connections are not described and their order is not always logical (introducing EC and only later the less suitable alternatives). L36-44 is good example, where I am not sure what the message is: ET is both mass and energy; the composition of terrestrial ET or the different observational techniques. Some of these messages also come back in the next paragraph. In this case, I would reorder the paragraphs to separate the explanation of the ET process and the observations. This is not the only paragraph where this issue occurs. The same can be seen in the abstract, which is a summation of facts that are not actively linked. This leaves it up to the reader to fill in the blanks.

The structure of the manuscript stays a limiting factor in its understanding beyond the introduction. Apart from links between paragraphs, short phrases explaining what will be said in a section have the potential to improve the manuscript. I will comment on the textual side in more detail, once the structure has been improved.

We agree that there was room for improvement in the organization of the text. We have restructured the manuscript to link the introduction and the discussion to the results, and to the assumptions that terrestrial ET comes mostly from plant transpiration and soil evaporation, without considering latent heat fluxes from anthropogenic sources such as car combustion or house heating. We also better explained our intention to develop a method capable of providing ET maps for mitigating the urban heat island effect and droughts by better managing vegetation in cities, which justifies the focus on plant transpiration and soil evaporation.

**Missing literature**

An elaborate overview of models is given (hydrological, surface energy models, L70-87). It feels like the models designed for the cities are overlooked, while they would be appropriate models. Grimmond et al., (2011) evaluated this type of models concluding ET is the most poorly modelled flux making the analysis of the authors even more interesting. The authors state (L94): "Most ET modelling approaches assume a landscape of homogeneous vegetation without anthropogenic elements." By which they omit a significant part of the urban climate research field. This is also missing in the discussion, where SVAT and hydrological models are discussed, but no urban climate models.

Thank you for the suggestions. We added specialised urban models such as Urban Climate Models (UCM) and Urban Land-Surface Models (ULSM), such as SUEWS, SWMM-UrbanEVA, PALM-4U and UT&C in the Introduction. Their pros and cons data requirements and model accuracy for ET/LE are presented in the introduction and later the SUEWS and UT&C model accuracy reported in the literature are compared to our results in the Discussion.

**Choice for SCOPE**

This brings me to a related point and my main concern, which is the missing argumentation for the choice for SCOPE. The inclusion of vegetation and soil in the model is interesting, but it is not developed for the city. Other models UT&C (Meili et al., 2020) and SUEWS (Järvi et al., 2011) also include vegetation and soil, but are designed for urban areas already. Given the authors argue that the model is not suitable for urban areas (L229-230), why is SCOPE adapted for urban areas instead of using remote sensed input for a readily adapted model? SCOPE is not the most appropriate model for the study at hand, and the author should consider redoing the analysis with a model designed for urban areas.

As mentioned before, the intention is to develop a method capable of providing ET maps for mitigating the urban heat island effect and droughts by better managing vegetation in cities. Therefore, our focus is on soil evaporation and plant transpiration. A sophisticated Soil-Vegetation-Atmosphere Transfer (SVAT) model such as the SCOPE model (Soil Canopy Observation, Photochemistry and Energy fluxes) provides the necessary framework for our application and the prediction accuracy compatible with the state-of-the-art in urban models but with fewer model inputs.

For instance, SUEWS models have many non-ordinary inputs that are difficult to supply in a high temporal and spatial resolution (Järvi et al., 2011). Rafael et al. (2020) state that the availability of measured data is a limitation for applications. The estimations of LE may be critical output in most urban models, often showing a low accuracy, especially in dense urban areas. Rafael et al. (2020) applied the SUEWS model in two locations in Portugal, concluding that the performance of LE predictions in suburban areas was far better than the denser urban site (correlation 0.61 and 0.13, respectively). The statement is consistent with previous studies using two areas with different levels of urbanisation, conducted in the surroundings of London (R² 0.72 and 0.25) and Helsinki (correlation 0.79 and 0.44) (Karsisto et al., 2016; Ward et al., 2016).

Although the UT&C model is a very sophisticated and detailed urban model (i.e. urban canyon design), the accuracy is similar to the SUEWS models. UT&C models require even more complex parameters, including the wall's distance to the tree trunk (m), albedo and emissivity of walls, the thickness of walls and roofs, and volumetric heat capacity of impervious surfaces, roofs and walls. The R² reported for the three locations (Singapore, Melbourne and Phoenix) range from 0.50 to 0.62 (Meili et al., 2020). However, given that the model was developed and calibrated for these sites, the accuracy may be lower when transferred to a different location or period.

Our modelling approach also presents better accuracy for the suburban site ROTH (R² 0.82) than the build-up area TUCC (R² 0.47), similar to the SUEWS models. In general, the accuracy of the dense urban sites is lower than more vegetated areas, independent of the model approach. However, a specialised urban model should perform optimally in denser build-up areas as they were designed for such environments.

**Clarity of methodology**

While the actions taken reasonably well described, the methodology lacks explanation of the design choices. This raises questions for example regarding why data was left out and altered in a certain way. The effect of the choices should also be discussed. In the minor comments below, I indicate examples of choices for which the reasons are not clear. Also, for some parts references or explanations are missing, which is also indicated below.

To better explain the methodology, we have included more information about your assumptions, data processing, the model approach and the correction factor.

**Correction factor**

Defining the correction factor as the vegetation fraction creates an assumption that is not mentioned in the manuscript. By multiplying the predicted ET by the vegetation fraction it is assumed that this fraction produces all relevant ET. At the same time, literature shows impervious surfaces make a significant contribution to ET (Ramamurthy et al., 2014; Wouters et al., 2015). Therefore, this limits the conditions for which the presented model can be applied. In case of completely impervious area, the correction would lead to a permanently zero ET, which is not realistic.

As mentioned above, we refined the explanation of your assumptions and the correction factor. We have also included an in-depth discussion about the intercepted precipitation (lines 414 to 440) and

the impacts of considering complete impervious as zero ET, which models as SUEWS also do (excluding evaporation from interception loss).

In the discussion, the authors give a very complete framing to the results. The discussion also includes a lot of repetition and remarks that do not seem to have a purpose. Therefore, I think it can be shortened considerably. Implications of drawbacks and assumptions that are mentioned for the results are often not mentioned. The conclusion is very to the point, but should be more specific regarding amongst others the performance of the model.

We have restructured the Discussion completely, changing the four sections to a more concise three sections. The conclusion was also improved as suggested.

The figures properly support the text and show the results clearly. I especially liked Figure 6. Unfortunately, the image quality of the figures is poor. For some, the images are grainy and individual pixels can be seen. They are also not easy to read due to missing legends and titles for panels. The captions are very long with a lot of information on how to read the figures. Legends and subtitles could help to transfer some of this information to the figures themselves. In the minor comments I included remarks per figure.

All figures (but six) were remade improving resolution quality, readability and including legends to reduce the need to read the captions.

**Minor/technical comments**

Apart from these larger concerns regarding the methodology and reflection, I have some smaller points that could help improve the manuscript. Also, I wrote down a list of textual improvements, which is not meant to be an exhaustive overview. Since I think the writing needs major adjustments, I have not corrected the language in detail. The points are indicated with a line number.

L11-13: It sounds contradictory that most models assume a homogenous vegetated landscape, but at the same time they lack input parameters to describe the land surface.

We agree that is not exactly "land surface" but more detailed soil and vegetation properties than assume grassland or crop factor. It was rephrased, changing land surface to "lacking explicit input parameters to precisely describe vegetation and soil properties"

L22: It is unclear to me what the respectively refers to, is that the different sites?

We agree that "respectively" has no function in this case. It was excluded.

L24-25: I would like to see a more concrete suggestion for an application, instead of a very general "statement that fits under almost all urban climate research. What does this research specifically add?

We rephrased the novelty and the potential application to better highlight what the research adds. "This study aims to develop a robust and transferable method to map urban ET at any location in the city using a high-resolution spatiotemporal model that requires only freely available data inputs. The novelty is to provide a solution that combines the high temporal dynamic of ET in a vegetated environment with the spatial fragmentation in urban environments, producing a less computationally expensive but plausible ET product. We assume that terrestrial ET is mostly derived from plant transpiration and soil evaporation, considering these sources to be essential in mitigating the UHI and droughts by better managing green areas in the cities. We neglected interception loss from precipitation and latent heat fluxes from anthropogenic sources such as car combustion or house heating. These sources are not directly associated with ET's cooling effect and may mislead urban planning as they are likely inversely proportional to UHI and droughts. We propose a process-based

SVAT model (i.e. SCOPE) combined with a correction factor for urban environments based on vegetation fraction to derive hourly ET. The factor corrects the model bias due to impervious surfaces using vegetation fraction extracted by hourly footprints. The hourly predictions for an entire year (12 months, 24 hours, 8760 timestamps) were compared to reference ET derived from the Penman-Monteith equation and validated with flux tower measurements from two locations in Berlin, Germany. The study focuses on modelling with open data from standard meteorological stations and remote sensing products available for most medium and large cities of Europe, targeting transferability."

L30: I think the adverse health effects of the UHI are better described in other papers than Vulova et al., 2020. Either remove or replace this citation, since the other two are appropriate.

We have removed the mentioned citation.

L34: In my opinion, it is not substantiated here why ET us so important in urban areas, although I do fully agree with the statement itself.

It was changed to: "Although ET plays an essential role in planning more sustainable cities, studies in urban environments are rare and very localised due to the challenges of measuring and modelling evaporation in highly heterogeneous landscapes (Nouri et al., 2015)."

L63: A practical solution to what problem?

We rephrase to a better understanding. "As urban ET observations are rare, costly and available only for a few cities in the world, an alternative is to estimate ET using process-based or empirical models"

L101: Why is the choice here to use evaporation, while in the whole manuscript evapotranspiration is used?

We agree and replace it for ET.

L104-105: This sentence is not correct English.

Replaced for: "For instance, impervious areas are mainly static over a one-year interval, while characterising the weather conditions in an hourly resolution is desirable. Thus, a model that embedded all the interactions between atmospheric conditions, vegetation and soil properties, impervious fractions and anthropogenic heating would be mostly redundant in space or time for hourly ET estimation"

L115-118: The use of comma's would improve the readability of long sentences like this (or consider breaking them up).

We have broken it up: "Two sites in Germany's biggest city and capital, Berlin, were selected for this study because they are equipped with eddy flux towers."

L123: Omit "(mm)".

It was removed.

Figure 1: The color scale for the impervious fraction is hard to read due to the use of two colors that do not have a different meaning. It would also not be readable in black and white.

The colour scale was replaced by shades of grey to better represent the imperviousness

The buffers are drawn at 1500 m, but in literature 500 m is used more often (e.g. Coutts et al., 2007 and Hong et al., 2020).

We changed to a 1000m buffer. Despite the literature often referring to 500m, one of the towers is installed at 56 meters height, which generally increases the area of influence despite the footprint area being over 500 m in most of the atmospherical conditions. Some authors like F. Lindberg et al. (2018) present an even bigger source area (Environmental Modelling & Software 99 / 70-87).

It could be useful to use letters for the different panels to ease referring to parts of the figure.

Figure 1. Locations of the two sites with the respective (a) vegetation fraction (%), (b) impervious fraction (%) and (c) vegetation height (m) in the surroundings of the flux towers (d). The red dotted areas represent a buffer of 1000 m around the towers (red dot), while the red ellipses are examples of hourly footprints. The black dots on the Berlin map (c) refer to the DWD weather stations Tegel and Dahlem. The three land surface maps were extracted from the Berlin Digital Environmental Atlas (Senate Department for Urban Development and Housing, 2017; Senate Department for Urban Planning and the Environment, 2014).

L136-139: The second sentence repeats much of what is said in the first one (e.g. a sonic anemometer measures orthogonal wind components).

We have excluded the sentence "The towers are equipped with micrometeorological instruments to simultaneously measure water vapour density and orthogonal wind components, allowing for ET estimation."

L140: I would say this is part of a site description, not the data.

These sentences were transferred to the "study area" section.

L146: The authors state: "Negative values (condensation) were set to zero." I think this affects the results, while it is not based on anything physical. What are the effects on the results? Has an analysis without setting all these values to zero been performed.

As both modelled and observed ET was set to zero the effect on the results are minimal. For instance, in the SCOPE model scenario with DWD + RS data (corrected), the metrics are RMSE = 0.0256, R2 = 0.8207 and rBias = 0.0065, while preserving both negative values the accuracy is almost the same with RMSE = 0.0258, R2 = 0.8195 and rBias = 0.0120. The two main reasons to set to zero are (1) because we show the annual and monthly ET (mm) estimations, and the negative values would neutralise some positive values in the sum, and (2) because most of the available ET products provide values and scale starting at zero as condensation is not considered, for instance, MODIS Global Evapotranspiration Project (MOD16 - NASA/EOS)

The sentence was completed to state the choice. "Negative ET values (condensation) were set to zero as annual sums in millimetres will be provided and we are only interested in the amount of water released into the atmosphere by soil evaporation and plant transpiration processes."

L146-148: The used prepositions seem odd here.

Some prepositions were replaced and the sentence restructured: "The entire year of 2019, including winter and nighttime, was selected as there are EC observations simultaneously available for both towers in 2019. "

L151-153: The choice to leave out the 4 hours directly after precipitation will cause a problem, since at this time ET is especially high in cities (Ramamurthy et al., 2014). What is the influence on the conclusions?

The ET was predicted for all hourly timestamps in 2019, including during and after rain. However, the validation could not occur in these periods as the observed ET from EC tower data is missing or not reliable (Kotthaus and Grimmond, 2014). We agree that ET from wet surfaces are even higher in cities as we have stated in the discussion, "According to Ramamurthy and Bou-Zeid (2014), wet impervious surfaces evaporate at higher rates than wet vegetation, as they often store more heat. They conclude that evaporation from wet impervious surfaces such as concrete pavements, asphalt and building rooftops accounted for around 18 % of the LE and may last up to ten days, with the highest evaporation rates occurring 48 hours after a precipitation event. In our study, we have excluded data up to 4 hours after precipitation events from the validation dataset to assess model accuracy." However, we are not considering interception loss in our study, and we assume that this evaporation is the effect of intercepted precipitation rather than the result of soil evaporation and plant transpiration.

L155-157: What methodology was used to calculate the footprints?

The methodology was explained in lines 186 and 187 "The Kormann and Meixner (2001) analytical footprint model was applied using the R package "FREddyPro" (Xenakis, 2016)". To improve readability, this sentence was transferred to lines 193-194.

L173: This is not the first point the content of Table 1 is discussed.

It was moved to line 155.

L179: Why is linear interpolation performed if the differences are minor and irrelevant to the study?

The interpolation is performed to have a value for each timestamp to provide as model input and also to fill the time series gaps caused by clouds.

Table 1: This table summarizes a lot of information very nicely. It also feels like a lot of repetition from the text, which provides the opportunity to shorten the methodology.

We excluded from the texts "hourly time series of air temperature, air pressure, relative humidity, wind speed, wind direction, and precipitation (occurrence and volume)." and "The following measurements from both towers were used to calculate the footprints: wind speed (ws, m s$^{-1}$), wind direction (wd, degree), friction velocity (u*, m s$^{-1}$), Obukhov length (L, m) and northward wind (v_var, m$^2$ s$^{-2}$) (Table 1)."

Rephrased: "The data from the meteorological stations, Tegel (~5 km from TUCC) and Berlin-Dahlem (~1 km from ROTH), were used as model inputs (Table 1)."

L188-190: Repetition of L155-157.

Excluded. "The footprints vary according to atmospheric stability (Monin-Obukhov stability parameter) and wind components (direction, speed, cross-stream, friction velocity) interacting with the surface roughness around the tower (Kljun et al., 2002; Kormann and Meixner, 2001)."

L196-197: This method is not clear to me, what pixels are used and which are not?

We have restructured the sentences for a better understanding. "The footprints were based on a regular grid of 10 m resolution with an extent (x, y) of 1000 m from the tower locations (fetch size). For

each grid pixel, the probability that the source area belongs to the flux measurements influence zone was calculated for every hour (Schmid and Oke, 1990). These grids of probabilities, excluding pixels outside of 90 % of the footprint likelihood, were multiplied to the raster of the surface property (e.g. vegetation height) to extract average values for each timestamp of 2019."

L198: What was the original resolution of this dataset? And is this one map for the entire year?

The original resolution of the vegetation height dataset is 1m resolution. The vegetation fraction is based on polygons containing block and block sub-areas, but also includes roads, street trees and rail tracks. The polygons were converted to a 10m resolution raster to extract the footprints. The maps are from a specific time (2017), so indeed one map for the entire year. However, as 30 min footprints were extracted, the model inputs vegetation height and vegetation fraction very hourly. We assume that these variables are nearly static over a year. "All the layers of GIS maps were converted to a raster with 10 meters resolution and resampled to the footprint grid of each tower to extract the average surface properties per timestamp. The raster layers of each land surface were then multiplied by a footprint raster, and the resulting pixel values were summed to obtain the weighted averages for each site and timestamp."

Figure 2: Please include a legend indicating the meaning of colors and line style (it is now only in the caption).

The colours and line types for each plot are included now.

L221-223: Given that this is the definition of benchmarking (Best et al., 2015), it would be good to use that term.

The word baseline was replaced by benchmarking.

L227: This sentence is not correct English.

We rephrased the sentence to "SCOPE is an ensemble model approach, combining one-dimensional bidirectional turbid medium radiative transfer, micrometeorology and plant physiology (van der Tol et al., 2009). "

L229: In my view, the model will not provide but require a lot of input, which in turn can be positive, but also has the potential to limit the usability of a model.

We rephrased the sentence as "This configuration allows SCOPE to account for a wide range of surface-atmosphere interactions, requiring different model inputs according to the target outputs."

Note: Despite SCOPE including more than 60 options of parameters, it allows users to run the model using all the inputs as default constants. Thus, none of the parameters must be changed or varied. But of course, to have meaningful results important inputs for the target output should be customized to the location and timestamp. The description of SCOPE was updated with more information about the model.

L233: Replace "group" with "groups".

Thank you. Corrected.

L237: What are these other parameters at the end of the line? The same as the ones mentioned earlier?

There are more than sixty parameters, divided into soil, leaf, canopy, biochemical parameters besides the meteorological ones. For instance, some examples of SCOPE parameters are soil brightness (BSM), chlorophyll content (Cab), leaf water content (Cw), CO2 concentration, stomatal conductance, soil and canopy resistance to evaporation, and photosynthetic parameters (e.g. vcmax).

Table 2: The unit of air pressure is stated as ppm, but this seems odd to me since this is not the standard unit.

We replaced it for hpa. Ppm was a mistake as it is the unit of the CO2 variable tested in some preliminary results.

L255: The direction of bias would be good to include.

Included (i.e. overestimated).

L257-259: Is the vegetation fraction the correction factor? It seems logical, but it is not stated explicitly.

The sentence was slightly changed to be more explicit that the correction factor is an hourly estimation of the vegetation fraction. "In order to correct the ET predictions according to the surface characteristics of each site, we use the extracted vegetation fraction average from the footprints per timestamp to subtract the ET estimated in impervious areas with a 10 meters resolution product. The correction factor for urban environments is a relative value that varies from 0 to 1, where zero means completely impervious and one fully vegetated."

Note: If the impervious fraction is used the results are very similar. The factor was also tested, correcting the soil and canopy LE separately using impervious fraction (complementary to vegetation fraction) and vegetation fraction respectively. But we opted for the vegetation fraction for simplification as the results were very similar.

L259-260: This is stated more elaborately in L263-269.

The sentence was excluded. "Then, corrected ET model predictions are compared with the observed (hourly aggregated) ET from the flux tower to assess model accuracy."

L263: All available ET values observed before or after the filtering described earlier?

Hourly ET values were predicted for the timestamps of the year (8760), but the validation was performed using the cleaned/filtered EC data (around 60%). The remaining ET values were missing values or had no reliable observations.

L266: What is the definition of rBias?

It is the sum of the differences between predicted and observed values of each timestamp relatively to the total ET observed in the period.

L272-273: Repetition of L75-76.

It was excluded.

L274-275: The reference is to only one panel of the figure (3a), while results for any combination are stated. What are all the possible combinations and are these all shown?

The sentence was slightly changed. The sentence refers to the combination of 2 by 2 tower location and station location (Dahlem-ROTH, Tegel-TUCC, Tegel-ROTH, Dahlem-TUCC, ROTH-TUCC, Tegel-Dahlem). "The results (Fig. 3a) show that there is a strong relationship between the ETo calculated using data from flux towers (x-axis) and data from DWD stations (y-axis), but also between the locations using the same data. For any of the six combinations of ETo pairs, the coefficient of correlation is at least 0.96 (not shown)."

```
> round(cor(ETo.DWD.ROTH, ETo.EC.ROTH), 3)      [1] 0.976
> round(cor(ETo.DWD.TUCC, ETo.EC.TUCC), 3)      [1] 0.966
> round(cor(ETo.DWD.TUCC, ETo.EC.ROTH), 3)      [1] 0.976
> round(cor(ETo.DWD.ROTH, ETo.EC.TUCC), 3)      [1] 0.964
> round(cor(ETo.EC.ROTH, ETo.EC.TUCC), 3)       [1] 0.982
> round(cor(ETo.DWD.TUCC, ETo.DWD.ROTH), 3)  [1] 0.996
```

L277-278: A very interesting result that could be highlighted more given its wider potential for application.

We agree that this is one of the highlights of the method. As only open data was used, it has a great potential to be generalized to other places without EC flux towers. We change the lines to the following: "Therefore, we use only publicly available meteorological model inputs to predict ET, completely independent from the measurements of the two towers. As meteorological variables and vegetation fractions are available for most medium and large cities of Europe, there is a great potential for the methodology to be transferred for other locations based on the promising results shown for the two EC towers in Berlin."

Figure 3: Panel a has a y-label stating DWD data and the others simply state DWD. Are these the closest DWD stations to the flux towers or were multiple combinations tested?

In panel a), plotted ETo is calculated from the two closest DWD stations against the ETo calculated from the towers' data. Now the figure states "DWD data" for all predicted ETo using DWD data.

On top of that, panel b-c show ET_0, but in d-f this is replaced by predicted ET. $ET_0$ is in this manuscript also treated as a prediction.

Now the figure states "predicted" also for ETo from DWD or EC data.

A minor detail is the inconsistency of the capitals in the x-labels.

Thank you, corrected.

In addition, panel a does not show the observed ET on the x-axis. Panels b-f tell a story together (evolving precision using different approaches), but panel a is different and as shown now feels as if it should be a separate figure.

We agree that panel "a" is different from the others, but for the sake of saving space and having an even number of panels to compose the figure, we decide to group them together.

This figure may benefit from titles for the subpanels to in one glance see the differences between the plots.

We included titles for each panel.

L285: This sentence contradicts itself. If atmospheric conditions are the main driver, why is it mostly depends on the land surface?

We rephrased to be clearer about our claim. "Although atmospheric conditions and water availability mainly drive the temporal variability of ET, the spatial variability, which determines the volume of ET, depends primarily on the land surface characteristics."

L286-287: Repetition.

Deleted and the following sentences were adjusted. "The models clearly overestimate ET in highly fragmented landscapes with impervious surfaces, as shown in Figure 3b. The difference between the two towers emphasises the dependence on the vegetation fraction."

L288: Please add panels to the figure reference, since some are corrected for this assumption.

Done. Please see above.

Table 3: The caption states that the highest precision and lowest bias are highlighted in bold, but this is not shown in the table.

It was corrected.

The names of the scenarios may cause confusion, since $ET_0$ is both a model approach and an input scenario. For example, in L303, it is bit clear without checking the values in the table what is the corrected $ET_0$ prediction.

It was included Penman-Monteith in the sentence to clarify and revised throughout the manuscript. "The corrected ETo prediction from Penman-Monteith, which initially presents a rBias of 1.57 and 3.83,…"

L317: What is meant with the word "selected" in front of SCOPE?

In this case, it is the SCOPE_RS model. We tested some different model settings, for instance, applying or not applying the SCOPE correction for vcmax using temperature. But since all SCOPE models in the different scenarios and locations used the same settings, no model selection was performed in this context. "Selected" was replaced by "SCOPE_RS".

L320: This suggest SCOPE produces output for specific sources of ET, which would be very interesting. Is this correct?

SCOPE provides latent heat flux for the soil, the canopy and total. We have tested correcting differently for soil and canopy and for daytime and nighttime. The inner-city site (TUCC) presented improvements in accuracy while the ROTH site showed no significant change. For the sake of simplification, as this is the first publication with this approach, we decided to show only the total LE.

L324: Panels 4c and d are referenced before a and b.

As all panels share the x-axis, we think it is better to keep the x-axis values near the ET prediction panel. But panels c and d that refer to ET are described before a and b as they are more important.

330: The observed ET may be influenced by an anthropogenic moisture flux. How relevant is this at the study sites?

We believe that the anthropogenic moisture flux is more important in winter and nighttime at TUCC. But we do not have data to assess how important it is. However, we believe that most of this disagreement between predicted and observed ET is caused by evaporation from interception loss from precipitation, especially on the TUCC site.

L332-333: Is ET prediction in SCOPE linked to the water availability?

Yes, it is possible to use soil moisture content (SMC) to limit the water availability, but as we have SMC data only for one location (ROTH), we have decided not to use this parameter. In some tests for ROTH, the use of SMC helped to reduce the prediction overestimation in the driest month (April, the only month clearly overestimated). However, as on very dry days institutions and citizens often irrigate their gardens, we then noticed a tendency of underestimation in very dry conditions. Therefore, this relationship has to be further explored in future works.

Figure 4: A very insightful figure, but for panel a and b a legend is missing.

The legend was included.

Figure 5: In order to clarify the link between the error and the precipitation, showing them in the same graph (reducing the panel count to 2) makes comparison easier. What is the meaning of the dotted lines in panel b?

They are now combined in two panels. The dotted line was the average for both locations.

L360-363: Repetition.

It was removed and the next sentence was rephrased. "As 42 % of the hourly ET observations were missing values, we performed the MDS gap-filling method to estimate monthly or yearly observed ET values. 336 mm/year was estimated for the ROTH site, representing 66 % of the observed annual precipitation (Fig 6)."

L370: In order to value the comparison of the study at hand and the on done by the Senate Department for Urban Planning and the Environment, some background of that study should be discussed.

We included the following sentence to better define the compared model. "We also compared our approach with the hydrological water balance model (ABIMO 3.2), which models and maps evaporation from precipitation for Berlin available in the study "Surface runoff, percolation, total runoff and evaporation from precipitation" (Senate Department for Urban Planning and the Environment, 2019). This model requires approximately twenty-five data inputs for almost 25,000 single sections of the city (blocks, streets and other features), providing a detailed spatial resolution but only an annual temporal resolution which is not updated every year."

L375: I miss a comparison with the results of the work from the authors.

A comparison was included and others modelling approach was included. "For the block where the two EC towers are installed, the evaporation from precipitation was reported as 344 mm/year at ROTH and 196 mm/year at the TUCC site. When considering the average footprint of each tower, the annual values of the Berlin Environmental Atlas reduce to 266 mm at ROTH and 165 mm at TUCC. Our approach estimated 330 mm and 151 mm, respectively, while the EC observations (gap-filled with MDS) were 336 mm and 188 mm." From lines 519 to 548 the SUEWS and UT&C model results in the literature were compared with our results and approach.

L378-379: To what study is referred here?

It is referred to the Berlin Environmental Atlas study, which was referenced in the text. "Compared with the 2013 edition, evaporation estimated by the ABIMO model has decreased due to the increase of impervious surfaces and the expansion of drainage systems."

L406-408: Repetition.

It was removed.

L409-415: The message of this paragraph is not clear to me.

It was rephrased to "The effect of surface heterogeneity in the horizontal direction, typical in an urban environment, is not addressed by (1D) SCOPE or Penman-Monteith-based models. However, accounting for surface-atmosphere interactions in vegetated fractions with a SCOPE model combined with high-resolution land cover to mask the impervious areas makes it possible to predict ET accurately in urban environments."

L417-427: Repetition of the introduction.

It is now mostly excluded, keeping only the following sentence to introduce the drawbacks: "The EC system used in this study is one of the most suitable approaches for deriving observed terrestrial ET, especially in urban areas (Foltýnová et al., 2020; Nouri et al., 2013)."

L435-441: Since none of these observation methods is used, why is this relevant for the study at hand?

Excluded.

L453-455: I agree with the authors that the interception loss/evaporation will be important in urban areas, but the link to anthropogenic heat fluxes should be explained.

It was rephrased to explain the relationship with anthropogenic heat fluxes. "The model underestimation occurs mainly at night and winter, which makes us conclude that direct anthropogenic heat sources have a minor contribution to LE during the spring and summer. However, during winter, neither moisture nor the cooling effect capacity of ET is important in this part of the globe.", and "For the denser built-up site (TUCC), the lower accuracy and the relative underestimation (-0.13 of bias) in comparison to ROTH could mostly be attributed to interception loss combined with higher land surface temperature caused by anthropogenic heating (Fig. 5b)."

L458-462: To my understanding, removing these 4 hours would decrease the value of the modelling approach for water balance estimation, as the interception evapotranspiration is not (completely) take into account. As is also stated in the manuscript, but I do not understand why as a consequence these 4 hours are excluded from analysis.

As explained before, we removed up to 4 hours after precipitation from the EC data as the observations were not reliable, affecting only the model validation. We have estimated these hours using SCOPE and ETo. However, we acknowledge that as we did not take the interception loss into consideration, the ET estimated from soil evaporation and plant transpiration may be smaller than the evaporation from wet surfaces, as the radiation on a cloudy day can limit the ET calculated by SCOPE or ETo considerably. A comprehensive discussion about it was included from lines 414 to 440.

L462: There is always a dew point temperature.

It was rephrased. "Furthermore, in winter, condensation generates wet surfaces at night (cars, windows, roads, metal roofs), which evaporate again during the day, increasing the ET measured by the EC method, similarly to interception loss."

L462-464: I do not see why interception loss is comparable to water that is condensed and evaporates again or what the relevance is that it is never taken up by the soil or a tree. Additionally, I

doubt whether the water that evaporates from the mentioned wet surfaces has been detected by the EC systems since they are installed high above the surface.

The excerpt "without being uptake by the soil or trees" was taken out to simplify the idea. We compare interception loss and water that is condensed because both produce wet surfaces but are not captured by the model. In the TUCC site, the tower is installed only 10 meters above the roof on a completely impervious and mostly flat surface (with indentations forming puddles) which makes it likely that evaporation from this surface is detected. If evaporation occurs, then condensation and evaporation again, EC data may account for ET multiple times.

L466-467: To what bias is referred here and what is the source?

Examples of bias sources were included. "For instance, the underestimation in the ET predictions around winter and periods with higher precipitation could be an artefact of bias in EC measurements caused by water in the instrument. Ward et al. (2013) also indicate that LE measured by the EC method presents significantly higher values than modelled LE in the following hours after rainfall. EC measurements can also be unreliable during certain conditions such as non-steady-state or absence of well-developed turbulence."

L468-469: What are the implications of this difference? Are the models wrong or the observations or is something else going on?

Most probably, both models and observations are biased and imprecise in different ways and in different conditions. "Measured" ET presents a mix of random and systematic errors that can be up to 20% (Foken, 2008; Liang and Wang, 2020), but they are more prominent in certain conditions (e.g. wet surfaces and low turbulence). The implications are that the model will perform worse in certain conditions when you validate the model. However, sometimes the error is in the observations used for validation, while other times the error is in model inputs (e.g. SMC without considering irrigation) or in the concept used (no interception loss is considered). But in the calibration process, if you seek global accuracy you may increase the bias of the model to fit the observed ET in conditions where the model was closer to reality than the EC measurements. The following sentence was added, "Both predictions and observations present a certain level of bias and imprecision (random and systematic errors) that behave differently according to the environmental conditions and model calibration. Therefore, when seeking global model accuracy, one may increase the bias to fit the observed ET better in general, affecting predictions in other conditions in which the model could be closer to reality than the EC measurement. A better approach would be to calibrate the model separately for different conditions."

L470-471: If this correction is common practice, why is it not applied?

This sentence was removed. There are many pre-processing approaches to correct different kinds of bias (which can increase others), but it is not the goal of this study to test or compare all possible corrections of EC data.

L473-492: There are a considerable number of interesting and noteworthy statements in here, but I miss the connections.

This section was reformulated.

L494-501: In my opinion, this is a very important outcome of the study. It paves the way for the prediction of ET for cities without highly specialized observation equipment, such as EC. It deserves a spot light in the conclusion.

The conclusion and the aim of the study (introduction) were adapted to highlight these points.

L506-508: I was surprised by the clear daily cycle in the composition of the footprint. I assume this is related to the daily cycle of the wind direction and strength, but a short background on this cycle would help to interpret the results.

An explanation was included. "These differences in footprint size across the day are affected by alterations in atmospheric stability and wind speed, which, combined with the vegetation-impervious composition in the tower surroundings, determine the vegetation fraction in the zone of influence."

L511-512: Has this been tested for the approach in this study?

It was tested, showing good potential. But as there are no structured results to show, we have decided to remove them. Alternatively, a simpler footprint derived from wind speed and direction freely available from meteorological stations could be applied without losing significant accuracy (see Quanz (2018)).

L529: What is not comparable in what terms?

This state was excluded in the reformulation of the Discussion sections.

L542: I value the inclusion of a future perspective. However, I think in the previous subsections there are already parts of this perspective. It would help to keep them together. Also, part of the section itself are not a future perspective, but an evaluation of this study and how it was adapted to its goals.

This section was excluded from and part of the discussion moved to other sections.

L569: Please further specify "land surface".

We replaced "land surface" with "vegetation fraction".

L544-545: The verbs switch time in this sentence, so it is not clear whether this is a general statement or one on this study.

It was rephrased. "Modelling SCOPE separately for each season may improve the accuracy as aerodynamic, photosynthetic, soil, and canopy constants could be better specified for these periods."

Figure A1: This figure shows the data for April are missing to a great extent, but more nighttime observations seem to be there. Could this partly explain the poor performance during this month?

Well observed; this may help to explain the overestimation in April at the TUCC site.

L732-733: This link does not work.

It was replaced by

https://www.berlin.de/umweltatlas/_assets/wasser/wasserhaushalt/en-texte/ekd213.pdf

More information can be found at https://www.berlin.de/umweltatlas/en/water/water-balance/

*References*

Best, M. J., Abramowitz, G., Johnson, H. R., Pitman, A. J., Balsamo, G., Boone, A., Cuntz, M., Decharme, B., Dirmeyer, P. A., Dong, J., Ek, M., Guo, Z., Haverd, V., van Den Hurk, B. J. J., Nearing,

G. S., Pak, B., Peters-Lidard, C., Santanello, J. A., Stevens, L., & Vuichard, N. (2015). The plumbing of land surface models: Benchmarking model performance. *Journal of Hydrometeorology*, *16*(3), 1425–1442. https://doi.org/10.1175/JHM-D-14-0158.1

Coutts, A. M., Beringer, J., & Tapper, N. J. (2007). Characteristics influencing the variability of urban $CO_2$ fluxes in Melbourne, Australia. *Atmospheric Environment*, *41*(1), 51–62. https://doi.org/10.1016/j.atmosenv.2006.08.030

Grimmond, C. S. B., Blackett, M., Best, M. J., Baik, J. J., Belcher, S. E., Beringer, J., Bohnenstengel, S. I., Calmet, I., Chen, F., Coutts, A., & others. (2011). Initial results from Phase 2 of the international urban energy balance model comparison. *International Journal of Climatology*, *31*(2), 244–272.

Hong, J.-W., Lee, S.-D., Lee, K., & Hong, J. (2020). Seasonal variations in the surface energy and $CO_2$ flux over a high-rise, high-population, residential urban area in the East Asian monsoon region. *International Journal of Climatology*.

Järvi, L., Grimmond, C. S. B., & Christen, A. (2011). The Surface Urban Energy and Water Balance Scheme (SUEWS): Evaluation in Los Angeles and Vancouver. *Journal of Hydrology*, *411*(3–4), 219–237. https://doi.org/10.1016/j.jhydrol.2011.10.001

Meili, N., Manoli, G., Burlando, P., Bou-Zeid, E., Chow, W. T. L. L., Coutts, A. M., Daly, E., Nice, K. A., Roth, M., Tapper, N. J., Velasco, E., Vivoni, E. R., Fatichi, S., & others. (2020). An urban ecohydrological model to quantify the effect of vegetation on urban climate and hydrology (UT&C v1.0). *Geoscientific Model Development*, *13*(1), 335–362. https://doi.org/10.5194/gmd-13-335-2020

Ramamurthy, P., Bou-Zeid, E., Cha, Y., Park, S. S., Kim, K., Byeon, M., & Stow, C. A. (2014). Contribution of impervious surfaces to urban evaporation. *Water Resources Research*, *50*(4), 2889–2902.

Wouters, H., Demuzere, M., De Ridder, K., & van Lipzig, N. P. M. (2015). The impact of impervious water-storage parametrization on urban climate modelling. *Urban Climate*, *11*, 24–50.

**Reviewer 2**

**General comments**

The paper estimates ET from two urban neighborhoods in Berlin using a Penman-Monteith equation and the SCOPE model, postulates corrections to make these models work better for urban terrain, and then compares the models to observations from Eddy Covariance towers. For the SCOPE model, 3 levels of sophistication in imposing the input are tested.

The study unfortunately falls shorts in multiple aspects and would need significant improvements before it can be reconsidered for publication.

Thank you for the very thorough review and valuable suggestions for our study. We have addressed most of your concerns, revising the manuscript accordingly with your suggestions or explaining the reasons otherwise. The manuscript has passed through an extensive review and many parts of the manuscript were completely restructured.

**Specific comments**

1) What is the MAIN goal of the paper? The whole paper reads like an unstructured collection of discussions and results that do not seem to be motivated by any open scientific question or knowledge gap. It is obvious that ET models for natural terrain would not apply in cities, and that different models are needed. So that is not a new finding. There are multiple models that already handle urban ET with much more sophistication than SCOPE+correction, so that also is not a new aspect (listed as a novelty on lines 110-112). So what is novel about this paper?

We rewrote most of the introduction and discussion to highlight our main goals. The novelty of the study was described as "This study aims to develop a robust and transferable method to map urban ET at any location in the city using a high-resolution spatiotemporal model that requires only freely available data inputs. The novelty is to provide a solution that combines the high temporal dynamic of ET in a vegetated environment with the spatial fragmentation in urban environments, producing a less computationally expensive but plausible ET product. We assume that terrestrial ET is mostly derived from plant transpiration and soil evaporation, considering these sources to be essential in mitigating the UHI and droughts by better managing green areas in the cities. We neglected interception loss from precipitation and latent heat fluxes from anthropogenic sources such as car combustion or house heating. These sources are not directly associated with ET's cooling effect and may mislead urban planning as they are likely inversely proportional to UHI and droughts. We propose a process-based SVAT model (i.e. SCOPE) combined with a correction factor for urban environments based on vegetation fraction to derive hourly ET. The factor corrects the model bias due to impervious surfaces using vegetation fraction extracted by hourly footprints. The hourly predictions for an entire year (12 months, 24 hours, 8760 timestamps) were compared to reference ET derived from the Penman-Monteith equation and validated with flux tower measurements from two locations in Berlin, Germany. The study focuses on modelling with open data from standard meteorological stations and remote sensing products available for most medium and large cities of Europe, targeting transferability."

2) The introduction goes on and on about measuring ET generally, but the paper is about modeling ET in urban terrain. The introduction needs to be completely rewritten to focus on the particularities of the physics of urban ET and highlight the gaps that this paper fills. Also the current intro is very convoluted and the ideas are not ordered coherently.

We have excluded the parts about measuring ET and completely restructured the introduction.

3) The paper presents as one of its main novelties the correction method for SCOPE and ET0, but in fact this method and the resulting factors are never explain. From lines 104-105, it seems to just be the ratio of pervious-vegetated terrain for a given footprint. Is that so?

We have expanded the methodology section, explaining better the SCOPE model and the correction factor.

4) If this is the correction methods, then again what is the novelty. There are a number of models out there now that treat urban hydrology with a much higher degree of sophistication (including shading by building, in and out of canyon vegetation, ET from impervious areas ….). So this paper seems to only be relevant for users of SCOPE.

The novelty of this study is the solution that combines the temporal dynamic of ET in a vegetated environment with the spatially fragmented land cover in urban environments, providing a less computationally expensive but plausible ET product suitable for most cities in the world. The prediction accuracy (precision and bias) is compatible with the state-of-the-art in urban models and potentially more transferable and less demanding in terms of data availability and computational power.

Another important remark is that the study focuses on modelling with open data available for most medium and large cities of Europe. The intention is to present a model able to generalise for other locations in Berlin and also for other cities based on the promising results shown for the two EC tower locations. The focus of the study is terrestrial ET from plant transpiration and soil evaporation, without considering latent heat fluxes from anthropogenic sources such as car combustion or house heating. It also excludes interception loss. The intention is to develop a method capable of providing ET maps for mitigating the urban heat island effect and droughts by better managing vegetation in cities, which justifies the focus on plant transpiration and soil evaporation.

5) A potentially interest angle is the input to the model, where there are 3 tests with more detailed inputs. This could be an interesting question related to the value of land surface data in modeling urban ET, but it is skimmed over very superficially.

We have discussed more deeply the inclusion of different model inputs to characterize soil and vegetation properties.

6) Section 2.2.1 is difficult to understand. The corrections done to instantaneous data and averaged data are mixed and explained poorly. It makes it look like the authors used the EC package but did not really understand it. For example

The section was restructured to better describe the procedure and corrections performed on the EC data. All the EC data pre-processing (raw or instantaneous data) was performed by the Climatology Department of Technische Universität Berlin (TUB), which operates the flux towers. We performed the cleaning of the averaged data such as applying quality flags, thresholds and extreme values filtering. "The two eddy covariance (EC) flux towers are operated by the Chair of Climatology at the Technische Universität Berlin (TUB) as part of the Urban Climate Observatory (UCO) Berlin (Scherer et al., 2019; Vulova et al., 2021). The EC measurement system is based on an open-path gas analyser and a three-dimensional sonic anemometer-thermometer (IRGASON, Campbell Scientific). The software EddyPro (Version 6.2.1) was used to derive turbulent fluxes of sensible and latent heat by processing the raw data sampled at 20 Hz. The pre-processing of raw data at 30-min intervals was performed as suggested by Vickers and Mahrt (1997), including physical threshold filtering, statistical screening and spikes elimination. The double rotation method was applied by EddyPro for the calculation of a local streamlined coordinate system as determined by the flow statistics over the 30-min averaging period. Furthermore, EC-data were corrected for air density and sonic temperature for humidity, high- and low-frequency spectral corrections (Moncrieff et al., 1997; Webb et al., 1980)."

the authors write "high- and low-frequency spectral corrections using double coordinate rotation" but double coordinate rotation does not correct for missed spectral bands, it corrects for sensor alignment…

These sentences were rephrased. "The pre-processing of raw data at 30-min intervals was performed as suggested by Vickers and Mahrt (1997), including physical threshold filtering, statistical screening and spikes elimination. The double rotation method was applied by EddyPro for the calculation of a local streamlined coordinate system as determined by the flow statistics over the 30-min averaging period. Furthermore, EC-data were corrected for air density and sonic temperature for humidity, high- and low-frequency spectral corrections (Moncrieff et al., 1997; Webb et al., 1980)."

When the authors write "observations six standard deviations (SD) greater than the average (de-spiking)," is that for raw data or averaged ET? De-spiking is a term used for raw data cleaning.

You are right about the de-spiking; the spike elimination procedure was performed using the raw data (20-Hz). The six standard deviation procedure was to eliminate extreme values (outliers) of the EC data already averaged to a 30-min. resolution. The section explaining this was rewritten. "The preprocessing of raw data at 30-min intervals was performed as suggested by Vickers and Mahrt (1997), including physical threshold filtering, statistical screening and **spikes elimination**.", and "The 30-minute values of latent heat flux (LE, W/m2) under the following conditions were excluded: (1) observations with flag quality higher than one (Foken, 2008); (2) values outside of the thresholds of -100 W/m2 and 500 W/m2; **(3) observations six standard deviations (SD) greater than the average (outliers),** and (4) measurements during precipitation or up to 4 hours after rain events. Items one to three were performed using functions from the R package "FREddyPro" (Xenakis, 2016). The wind directions 17°–35° at TUCC and 54°–72° at ROTH are susceptible to distortion due to the mounting setup of the instrument (wind coming from behind the tower)."

Why do they need big leaf to measure ET??â¼

We use a function of the R package "bigleaf" to calculate (not measure) ET because SCOPE provides only the estimations of latent heat flux (LE). As EC data provides both LE and ET, we used the same function to calculate ET from LE for both SCOPE and EC data to be compatible. The sentence was rephrased as "ET was calculated from LE as a function of air temperature using the "bigleaf" R package (Knauer et al., 2018) in order to use the same procedure for both observed and modelled LE from SCOPE."

Line 158: how much are "4993 and 5104" in % of data

The percentage is the complementary of the missing values (43% and 42%) or 4993 and 5104 out of 8760 timestamps in the year 2019, which is 57% and 58%. We kept the absolute because it is more important in this case to show that we are modelling around 5 thousand points per location. Original: "After pre-processing, from the 8760 timestamps, 43 % of the ROTH and 42 % of the TUCC data were missing. The remaining values of ET, 4993 and 5104 values respectively."

Do they remove data when the wind is coming from behind the tower? This is not mentioned

Included in the text. "The wind directions 17°–35° at TUCC and 54°–72° at ROTH are susceptible to distortion due to the mounting setup of the instrument (wind coming from behind the tower). However, as we are using a deterministic model that does not require training and the effect on the model accuracy for ET was insignificant, these observations were preserved."

7) Line 510: it seems that the impervious fraction changes during the course of the data. I presume this is imply related to wind angle and the footprint analysis, but the actual impervious surface are not dynamically changing like the LAI of vegetation for example.

You are right, the impervious/vegetation maps/data are fixed, but the fractions vary according to the timestamp because of the footprint (wind speed and direction, atmospheric stability). The explanation was made more explicit "Although the GIS data such as vegetation height and vegetation fraction are fixed for a specific time, the corresponding source area of the EC flux measurements (e.g. ET or LE) continually varies in shape, size and orientation. Therefore, areas of influence (footprints) were calculated every hour of 2019 for both towers to capture the spatiotemporal dynamics of the surface properties."

8) The later discussion sections are just a long discussion of so many ideas, things one can do, improvements and so on that do not seem directly relevant to the paper.

The discussion section was restructured completely, changing the four sections to a more concise and relevant three sections.

**Technical corrections**

1. "Whereas" is used multiple time including in the abstract in a wring way linguistically.
   These sentences were rephrased, removing whereas.
2. Line 36, mm is not a unit of mass and watt is not a unit of energy….
   The introduction section was restructured and this mention was excluded.
3. Table 1 row 4, is it 300 or 333 m as they mentioned earlier?
   The product is based on PROBA-V 333m data but the product name is LAI 300m because this was mentioned in the two different resolutions, but to keep consistent we will report 300m.
4. Line 328: it seems to be overestimating rather than underestimating.
   Figure 4 shows that from Jan-Mar and Sep-Dec the observed (black lines) are higher than the predicted by SCOPE corrected (green lines) for both sites. Therefore, the statement "The corrected SCOPE models exhibit the opposite behaviour, being more accurate around the spring-summer and underestimating otherwise." is correct from our understanding. We changed figure 4 to better show it in two panels.
5. Lines 345-346: the authors write "As both models are deterministic, temporal autocorrelation in the residuals is not an issue." I am not sure what this means.
   It was rephrased. "As both approaches are deterministic, there is no assumption of the independent and identical distribution residuals as in empirical models. However, temporal distribution in the residuals (autocorrelation) can help identify in which environmental conditions the precision and bias in predictions affect the overall accuracy."

---

## Referee Report (RR1)

**Review of revision of "Modelling hourly evapotranspiration in urban environments with SCOPE using open remote sensing and meteorological data"**

For this review, I first focused on the authors' response to the comments from the reviewers together with the tracked changes. After that, I reread the "clean" version of the manuscript to assess the quality. Beforehand, I want to express that the authors have done a great job and the quality of the manuscript has improved considerably. Below, I explain my positive view on the revisions. A few points of concern and others out of curiosity are left. These should not be hard to address.

*Response to comments*

The authors have taken the comments of the reviewers seriously and have taken the time to respond to all of them. I would like to compliment the authors on the improvement of their figures, which are now easier to read and understand (Figure 6 is especially very insightful). The overview of available models that are able to model urban evapotranspiration is expanded and now in my view includes all major model types completing the introduction. This inclusion also showcases the added value of SCOPE compared to the existing models needing less parameters as input.

I do agree with the authors that interception does not mitigate UHI, droughts or make cities more sustainable. However, in order to create a complete overview of the water balance and specifically the composition of the ET signal observed with EC this is an important process, as shown by the literature cited by the authors. This is thus important for the final claim of the paper at the end of the conclusion: "Therefore, our approach is well-suited to produce ET maps that are highly relevant to urban planning and climate change mitigation.".

The discussion has been shortened and its readability and relevance have improved. The authors successfully frame their findings and limitations. However, I think the discussion could be more to the point. The sections are a nice separation, but the headers are not informative and already show that it are more 6 than 3 sections, as all headers name 2 topics. More descriptive headers and using more, shorter sections should solve the latter issue. The discussion still mentions topics (e.g. EC in urban areas) that are not relevant for this particular manuscript. This change was very successful in the introduction, which is now a pleasure to read.

In the new version, the novelty of the research is clearly argued to lay in the transferability of the model to other cities. In case this is possible, it would indeed be a great addition to the existing knowledge. Theoretically, the model should indeed be more transferable to other cities, given the limited number of parameters needed. From these sites, I find it hard to deduce whether model is indeed transferable. Has this been tested or is this planned? It would be nice to test it on a completely different setting and see how it performs. I am also wondering whether all the open data applied in this method is widely available for cities around the globe.

*New manuscript*

The new manuscript reads more easily and I is possible to follow the reasoning of the authors over the course of the research. A clear aim is defined and in my opinion attained. This makes the whole research more accessible and showcases the progress this paper presents.

When rereading the manuscript, I wondered about particular effects of the urban area on the vegetation. Does SCOPE in any way account for the clothesline effect (Oke, 2002) and additional

water supply to vegetation next to impervious surfaces? Otherwise, this should result in a relevant bias, since the change in the fluxes of these effects are considerable.

Table 1: The last link does not work (related to vegetation fraction).

L328: It may be convenient to state the definition of rBIAS.

**References**

Oke, T. R. (2002). *Boundary layer climates*. Routledge.

---

## Author Response (AR2)

Point-to-point

We are glad to have received many insights and relevant comments from the reviewers and the editor. The review process has led to a considerable improvement in our manuscript. Based on the remaining feedback from the minor revision, we carefully adjusted the manuscript to address the last few comments. Below we present a point-by-point summary of all edits to address these suggestions.

Most of the changes were in made the Discussion by reorganizing the sections and giving more informative headers. Also, we included some of the limitations of our approach more clearly, as suggested.

The reviewer's and editor's comments are in orange colour, while in black are our answers and in red the alterations in the text found in the track changes file.

**Reviewer #1**

For this review, I first focused on the authors' response to the comments from the reviewers together with the tracked changes. After that, I reread the "clean" version of the manuscript to assess the quality. Beforehand, I want to express that the authors have done a great job and the quality of the manuscript has improved considerably. Below, I explain my positive view on the revisions. A few points of concern and others out of curiosity are left. These should not be hard to address.

**Response to comments**

The authors have taken the comments of the reviewers seriously and have taken the time to respond to all of them. I would like to compliment the authors on the improvement of their figures, which are now easier to read and understand (Figure 6 is especially very insightful). The overview of available models that are able to model urban evapotranspiration is expanded and now in my view includes all major model types completing the introduction. This inclusion also showcases the added value of SCOPE compared to the existing models needing less parameters as input.

I do agree with the authors that interception does not mitigate UHI, droughts or make cities more sustainable. However, in order to create a complete overview of the water balance and specifically the composition of the ET signal observed with EC this is an important process, as shown by the literature cited by the authors. This is thus important for the final claim of the paper at the end of the conclusion: "Therefore, our approach is well-suited to produce ET maps that are highly relevant to urban planning and climate change mitigation.".

The discussion has been shortened and its readability and relevance have improved. The authors successfully frame their findings and limitations. However, I think the discussion could be more to the point. The sections are a nice separation, but the headers are not informative and already show that it are more 6 than 3 sections, as all headers name 2 topics. More descriptive headers and using more, shorter sections should solve the latter issue. The discussion still mentions topics (e.g. EC in urban areas) that are not relevant for this particular manuscript. This change was very successful in the introduction, which is now a pleasure to read.

In the new version, the novelty of the research is clearly argued to lay in the transferability of the model to other cities. In case this is possible, it would indeed be a great addition to the existing knowledge. Theoretically, the model should indeed be more transferable to other cities, given the limited number of parameters needed. From these sites, I find it hard to deduce whether model is indeed transferable. Has this been tested or is this planned? It would be nice to test it on a completely different setting and see how it performs. I am also wondering whether all the open data applied in this method is widely available for cities around the globe.

**Main points:**

1.1 Interception role in the water balance and in the EC signal contrast with conclusion: "Therefore, our approach is well-suited to produce ET maps that are highly relevant to urban planning and climate change mitigation.".

   The sentence was removed from the conclusion, and a similar but more specific phrase was placed in the new discussion section header, "Applications and limitations". The intention is to explain that the maps will not contain the complete water balance similar to the EC measurements but will be well-suited to applications to mitigate UHI.

1.2 The discussion could be more to the point and with more informative headers.

Indeed, following your suggestions and some of the editor concerns, we reorganized and renamed most of the headers to be more informative and specific. We also exclude some sentences to be more straight to the point.

1.3 Has the model transferability to other places been tested or planned?

So far, it was not tested in other places, but it is on the priority list for future works. This sentence was included: "The mapping for the entire city, the inclusion of different levels of imperviousness, a correction for intercepted precipitation, and the assessment of the model transferability in different locations will be explored in future works."

**New manuscript**

The new manuscript reads more easily and I is possible to follow the reasoning of the authors over the course of the research. A clear aim is defined and in my opinion attained. This makes the whole research more accessible and showcases the progress this paper presents.

When rereading the manuscript, I wondered about particular effects of the urban area on the vegetation. Does SCOPE in any way account for the clothesline effect (Oke, 2002) and additional water supply to vegetation next to impervious surfaces? Otherwise, this should result in a relevant bias, since the change in the fluxes of these effects are considerable.

References Oke, T. R. (2002). Boundary layer climates. Routledge.

The SCOPE calculates the fluxes over a homogeneous landscape (horizontally), and despite
the model accounting for the sunlit leaves (edges of the canopy), the edge effect of a vegetated area with another surface is not included. Also, as the correction and validation are performed by footprints that vary constantly and are based on raster data, it is not possible to account for edges in this approach. An object-based approach or canyon design may be needed to account for the clothesline effect, which is very complicated in a (vertical and horizontal) heterogeneous urban area.

Table 1: The last link does not work (related to vegetation fraction).

A new link was included (https://www.berlin.de/umweltatlas/en/biotopes/green-volume/) as the previous one was the direct link to download the WFS vector file from the ftp address, and it would not work directly on the browsers.

L328: It may be convenient to state the definition of rBIAS.

The definition was included: "The rBias is calculated as the sum of the differences between predicted and observed values of each timestamp relatively to the total ET observed in the period."

**Editor #2**

As you can see, referee #1 is generally pleased with the changes, and only has some minor comments at this point. These will need to be addressed when submitting a revised version. I have also evaluated the revised version myself, and I generally agree with the assessment of referee #1. The manuscript reads well, and most, if not all, of the issues identified in the open discussion have been adequately addressed. What I miss in the current version is a better and more critical discussion on the limitations of the approach. Under what conditions would, for instance, the assumption of zero ET from non-vegetated areas no longer be justified? Berlin is rather dry, and it seems likely that your approach would break down in wetter climates where rainy days are the norm, and ET from paved surfaces can no longer be neglected. Also you might want to propose a future, more stringent, test at another location where the vegetated fraction varies strongly with wind direction and speed. Only in this way it will become clear if the footprint modeling adds any information.

2.1 discussion on the limitations of the approach.

To accommodate a more specific section to discuss the limitation of our approach, we create a new header, "4.5 Applications and limitations", and have included new content discussing the main limitations of the approach (below).

"Limitations of our approach related to the neglect of the intercepted precipitation may occur when applying the model in very wet places where rainy conditions are predominant throughout the year. The proposed approach is not able to estimate the complete water balance similar to the EC measurements, which include the interception and anthropogenic sources of evaporation. However, our approach presents higher accuracy with fewer inputs compared to well-known models for urban ET that can be applied in high spatial and temporal resolutions. The latent heat flux estimate from SCOPE can be separated into soil LE and canopy LE, which allows incorporating different levels of imperviousness in the correction factor to overcome our prior assumption of no evaporation from (dry) paved surfaces. Maps derived from our approach are well-suited to support local governments in mitigating UHI effects during extreme summer temperatures as the neglected sources affect winter predictions more. The mapping for the entire city, the inclusion of different levels of imperviousness, a correction for intercepted precipitation, and the assessment of the model transferability in different locations will be explored in future works."

2.2 future tests with other locations where the vegetated fraction vary strongly with wind direction and speed and the role of the footprint modelling.

In the model validation discussion section (4.2) a sentence was added to highlight the role of the footprint and the necessity to further studies with more locations to arrive to a conclusion.

"Further investigation at other locations is needed to conclude the role of the footprint modelling to the overall prediction accuracy where the vegetated fraction varies strongly with wind speed and direction. Otherwise, a simple buffer estimation could be performed instead. While in the ROTH site, the correction using the vegetation fraction from footprints improves the model accuracy compared to a buffer, in the case of the TUCC site using a buffer of 500 metres presents slightly better accuracy than using footprints. This occurs since ET shows a moderate correlation (0.35 and -0.44) with vegetation fraction and impervious fraction extracted from the footprints for the ROTH site but no significant correlation for the TUCC site. However, vegetation fractions can partially explain the difference between observed ET and reference ET (ETo) in spring and summertime, presenting a correlation of 0.44 for the TUCC site and 0.62 when both locations are analysed together. In summer at the ROTH site, the percentage of vegetation fraction increases during the day up to noon, while the impervious fraction presents the opposite behaviour (Fig. 4b), which may partially explain the better correlation."

In the method section "2.2.3 Remote sensing and GIS data" the following sentence was changed to make more clear that the variation in the vegetation franction and height came from the footprint modelling as the GIS data (maps) are from a specific period.

"Although the GIS data such as vegetation height and vegetation fraction maps are derived from a specific point in time, the corresponding source area (footprint) of the EC flux measurements (e.g. ET or LE) continually varies in shape, size and orientation. Therefore, these two inputs were extracted using footprints for both towers, varying hourly to capture the spatiotemporal dynamics of the surface properties."